# The Equivalence of Dynamic and Strategic Stability under Regularized Learning in Games

**Victor Boone**     **Panayotis Mertikopoulos**
Univ. Grenoble Alpes, CNRS, Inria, Grenoble INP, LIG, 38000 Grenoble, France
{victor.boone,panayotis.mertikopoulos}@univ-grenoble-alpes.fr

## Abstract

In this paper, we examine the long-run behavior of regularized, no-regret learning in finite games. A well-known result in the field states that the empirical frequencies of no-regret play converge to the game's set of coarse correlated equilibria; however, our understanding of how the players' *actual* strategies evolve over time is much more limited – and, in many cases, non-existent. This issue is exacerbated by a series of recent results showing that *only* strict Nash equilibria are stable and attracting under regularized learning, thus making the relation between learning and pointwise solution concepts particularly elusive. In lieu of this, we take a more general approach and instead seek to characterize the *setwise* rationality properties of the players' day-to-day play. To that end, we focus on one of the most stringent criteria of setwise strategic stability, namely that any unilateral deviation from the set in question incurs a cost to the deviator – a property known as *closedness under better replies* (club). In so doing, we obtain a far-reaching equivalence between strategic and dynamic stability: *a product of pure strategies is closed under better replies if and only if its span is stable and attracting under regularized learning.* In addition, we estimate the rate of convergence to such sets, and we show that methods based on entropic regularization (like the exponential weights algorithm) converge at a geometric rate, while projection-based methods converge within a *finite* number of iterations, even with bandit, payoff-based feedback.

## 1   Introduction

**Background.**    The question of whether players can learn to emulate rational behavior through repeated interactions has been one of the mainstays of non-cooperative game theory, and it has recently gained increased momentum owing to a surge of breakthrough applications to machine learning and data science, from online ad auctions to multi-agent reinforcement learning. Informally, this question can be stated as follows:

> *If each player follows an iterative procedure aiming to increase their individual payoff,*
> *does the players' long-run behavior converge to a rationally admissible state?*

A natural setting for studying this question is to assume that each player is following a no-regret algorithm, i.e., a policy which is asymptotically as good against a given sequence of payoff functions as the best fixed strategy in hindsight. In this framework, the link between learning and rationality is provided by a folk result which states that, under no-regret learning, the empirical frequency of play converges to the game's set of *coarse correlated equilibria* (CCE) – also known as the game's *Hannan set* [26]. This result has been of seminal importance to the field because no-regret play can be achieved via a wide class of "regularized learning" policies, as exemplified by the *"follow-the-regularized-leader"* (FTRL) family of algorithms [45, 46] and its variants – optimistic methods [15, 28, 41, 42, 47], HEDGE / EXP3 [4, 6, 10, 11], implicitly normalized forecasters [1, 3], etc.

All these policies have (at least) one thing in common: they seek to provide the tightest possible guarantees for each player's individual regret, thus accelerating convergence to the game's Hannan set. As such, in games where the marginalization of coarse correlated equilibria coincides with the game's Nash equilibria (like two-player zero-sum games), we obtain a positive equilibrium convergence guarantee: the long-run empirical frequency of play evolves "as if" the players were rational to begin with – i.e., as if they had full knowledge of the game, common knowledge of rationality, the ability to communicate this knowledge, etc.

On the other hand, in many other contexts, the marginals of Hannan-consistent correlated strategies may fail even the weakest axioms of rational behavior and rationalizability (such as the elimination of strictly dominated strategies). In particular, a well-known example of Viossat & Zapechelnyuk [49] (which we discuss in detail in Section 4) shows that it is possible to have *negative regret* for all time, but still employ *only strictly dominated strategies* throughout the entire horizon of play.

The reason for this disconnect is that no-regret play has significant predictive power for the empirical frequency of play – that is, the long-run empirical distribution of pure strategy *profiles* – but much less so for the players' day-to-day sequence of play – i.e., the evolution of the players' *actual* mixed strategies over time. In particular, even when the marginalization of the Hannan set is Nash, the actual trajectory of play may – and, in fact, often *does* – diverge away from the game's set of equilibria [16, 24, 34–36] or exhibits chaotic, unpredictable oscillations [12, 38]. Thus, especially in the context of regularized learning – where players learn *independently* from one another, with no common correlating device – the blanket guarantee of no-regret play may quickly become irrelevant, and even misleading, providing the veneer of rational behavior but not the substance.

Motivated by the above, our paper seeks to understand the rationality properties of the players' *actual* sequence of play under regularized learning, as encoded by the following question:

> *Which sets of mixed strategies are stable and attracting under regularized learning?*
> *Are these sets robust to strategic deviations? And, if so, is the converse also true?*

**Our contributions in the context of related work.** This question has attracted significant interest in the literature, especially in its pointwise version, namely: Which mixed strategy profiles are stable and attracting under regularized learning? Are the dynamics' stable states robust to unilateral deviations? And, if so, are these the only stable states of regularized learning?

In the related setting of population games, the answer to this question is sometimes referred to as the "folk theorem of evolutionary game theory" [14, 27, 51]. Somewhat informally, this theorem states that, under the replicator dynamics (the continuous-time analogue of the exponential / multiplicative weights algorithm, itself an archetypal regularized learning method), the following is true for *all* games: only Nash equilibria are (Lyapunov) stable, and a state is stable and attracting under the replicator dynamics if and only it is a strict Nash equilibrium of the underlying game [27, 51].[1]

In the context of regularized learning, [13, 20, 33] showed that a similar equivalence holds for the dynamics of FTRL in *continuous* time: a state is stable and attracting under the FTRL dynamics if and only if it is a strict Nash equilibrium. Subsequently, Giannou et al. [22, 23] extended this equivalence to an entire class of regularized learning schemes, with different types of feedback and/or update structures – from optimistic methods to algorithms run with bandit, payoff-based information. In all these cases, the same principle emerges: under regularized learning, *a state is asymptotically stable and attracting if and only if it is a strict Nash equilibrium.*

This is an important pointwise prediction but it does not cover cases where regularized learning algorithms do not converge to a point, but to a *set* (such as a limit cycle or other non-trivial attractor). In this case, the very definition of strategic stability is an intricate affair, and there are several definitions that come into play [7, 17, 21, 43]. The first such notion that we consider is that of "resilience to strategic deviations", namely that every unilateral deviation from the set under study is deterred by some other element thereof. Our first contribution in this direction is a universal guarantee to the effect that, with probability 1, in any game, and from any initial condition, *the long-run limit of any regularized learning algorithm is a resilient set.*

This result is significant in its universality, but the notion of resilience is not sufficiently strong to disallow irrational behavior – and, in fact, it is subject to similar shortcomings as Hannan consistency.

---

[1]Strictness here means that each player has a unique best response at equilibrium.

To account for this deficiency, we turn to a much more stringent criterion of setwise strategic stability, that of *closedness under better replies* (club). This notion, originally due to Ritzberger & Weibull [43], states that any deviation from a product of pure strategies is costly, and it is one of the strictest setwise refinements in game theory. In particular, it refines the notion of closedness under rational behavior (curb) [7], and it satisfies all the seminal strategic stability requirements of Kohlberg & Mertens [30], including robustness to strategic payoff perturbations.[2]

In this general context, we show that regularized learning enjoys a striking relation with club sets: *A product of pure strategies is closed under better replies if and only if its span is stable and attracting under regularized learning.* In fact, we show that this equivalence can be refined to sets that are *minimally* closed under better replies (in the sense that they do not contain a strictly smaller closed under better replies (club) set): a product of puer strategies is minimally club (m-club) if and only if its span is irreducibly stable and attracting (in that it does not contain a smaller asymptotically stable span of strategies). Finally, we also estimate the rate of convergence to club sets, and we establish convergence at a geometric rate for entropically regularized methods – like HEDGE and EXP3 – and in a *finite number* of iterations under projection-based methods.

In light of the above, our results can be seen both as a far-reaching setwise generalization of the folk theorem of evolutionary game theory, as well as a bona fide algorithmic analogue of a precursor result for the replicator dynamics, originally due to Ritzberger & Weibull [43]. Importantly, our analysis covers several different update structures – "vanilla" regularized methods, but also their optimistic variants – as well as a wide range of information models – from full payoff information to bandit, payoff-based feedback.

## 2 Preliminaries

We start by recalling some basics from game theory, roughly following the classical treatise of Fudenberg & Tirole [21]. First, a *finite game in normal form* consists of (*i*) a finite set of *players* $i \in \mathcal{N} \equiv \{1, \ldots, N\}$; (*ii*) a finite set of *actions* – or *pure strategies* – $\mathcal{A}_i$ per player $i \in \mathcal{N}$; and (*iii*) an ensemble of *payoff functions* $u_i \colon \prod_j \mathcal{A}_j \to \mathbb{R}$, each determining the reward $u_i(\alpha)$ of player $i \in \mathcal{N}$ in a given *action profile* $\alpha = (\alpha_1, \ldots, \alpha_N)$. Collectively, we will write $\mathcal{A} = \prod_j \mathcal{A}_j$ for the game's *action space* and $\Gamma \equiv \Gamma(\mathcal{N}, \mathcal{A}, u)$ for the game with primitives as above.

During play, each player $i \in \mathcal{N}$ may randomize their choice of action by playing a *mixed strategy*, i.e., a probability distribution $x_i \in \mathcal{X}_i := \Delta(\mathcal{A}_i)$ over $\mathcal{A}_i$ that selects $\alpha_i \in \mathcal{A}_i$ with probability $x_{i\alpha_i}$. To lighten notation, we identify $\alpha_i \in \mathcal{A}_i$ with the mixed strategy that assigns all weight to $\alpha_i$ (thus justifying the terminology "pure strategies"). Then, writing $x = (x_i)_{i \in \mathcal{N}}$ for the players' *strategy profile* and $\mathcal{X} = \prod_i \mathcal{X}_i$ for the game's *strategy space*, the players' payoff functions may be extended to all of $\mathcal{X}$ by setting

$$u_i(x) := \mathbb{E}_{\alpha \sim x}[u_i(\alpha)] = \sum_{\alpha \in \mathcal{A}} u_i(\alpha) x_\alpha \tag{1}$$

where, in a slight abuse of notation, we write $x_\alpha$ for the joint probability of playing $\alpha \in \mathcal{A}$ under $x$, i.e., $x_\alpha = \prod_i x_{i\alpha_i}$. This randomized framework will be referred to as the *mixed extension* of $\Gamma$ and we will denote it by $\Delta(\Gamma)$.

For concision, we will also write $(x_i; x_{-i}) = (x_1, \ldots, x_i, \ldots, x_N)$ for the strategy profile where player $i$ plays $x_i \in \mathcal{X}_i$ against the strategy profile $x_{-i} \in \prod_{j \neq i} \mathcal{X}_j$ of all other players (and likewise for pure strategies). In this notation, we also define each player's *mixed payoff vector* as

$$v_i(x) = (u_i(\alpha_i; x_{-i}))_{\alpha_i \in \mathcal{A}_i} \tag{2}$$

so the payoff to player $i \in \mathcal{N}$ under $x \in \mathcal{X}$ becomes

$$u_i(x) = \sum_{\alpha_i \in \mathcal{A}_i} u_i(\alpha_i; x_{-i}) x_{i\alpha_i} = \langle v_i(x), x_i \rangle. \tag{3}$$

Moving forward, the *best-response correspondence* of player $i \in \mathcal{N}$ is defined as the set-valued mapping $\mathrm{br}_i \colon \mathcal{X} \rightrightarrows \mathcal{X}_i$ given by

$$\mathrm{br}_i(x) = \arg\max_{x_i' \in \mathcal{X}_i} u_i(x_i'; x_{-i}) \quad \text{for all } x \in \mathcal{X}. \tag{4}$$

Extending this over all players, we will write $\mathrm{br} = \prod_i \mathrm{br}_i$ for the product correspondence $\mathrm{br}(x) = \mathrm{br}_1(x) \times \cdots \times \mathrm{br}_N(x)$, and we will say that $x^* \in \mathcal{X}$ is a *Nash equilibrium* (NE) if $x^* \in \mathrm{br}(x^*)$.

---

[2]Roughly speaking, robustness to strategic payoff perturbations means that the set under study remains stable even if the payoffs of the game are subject to small – but possibly adversarial – perturbations.

Equivalently, given that $u_i(x_i'; x_{-i})$ is linear in $x_i'$, we conclude that $x^*$ is a Nash equilibrium if and only if

$$u_i(x^*) \geq u_i(\alpha_i; x^*_{-i}) \quad \text{for all } \alpha_i \in \mathcal{A}_i \text{ and all } i \in \mathcal{N}. \tag{NE}$$

As a final point of note, if $x^*$ is a Nash equilibrium where each player has a unique best response – that is, $\mathrm{br}_i(x^*) = \{x^*_i\}$ for all $i \in \mathcal{N}$ – we will say that $x^*$ is *strict* because, in this case, $u_i(x^*) > u_i(x_i; x^*_{-i})$ for all $x_i \neq x^*_i$, $i \in \mathcal{N}$. An immediate consequence of this is that strict equilibria are *pure*, i.e., each $x^*_i$ is a pure strategy. Among Nash equilibria, strict equilibria are the only ones that are "structurally robust" (in the sense that they remain invariant to small perturbations of the underlying game), so they play a particularly important role in game theory.

## 3 Regularized learning in games

Throughout our paper, we will consider iterative decision processes that unfold as follows:

1. At each stage $t = 1, 2, \ldots$, every participating agent selects an action.
2. Agents receive a reward determined by their chosen actions and their individual payoff functions.
3. Based on this reward (or other feedback), the agents update their strategies and the process repeats.

In this online setting, a crucial requirement is the minimization of the players' *regret*, i.e., the difference between a player's cumulative payoff over time and the player's best possible strategy in hindsight. Formally, if the players' actions at each epoch $t = 1, 2, \ldots$ are collectively drawn by the probability distribution $z_t \in \Delta(\mathcal{A})$, the *regret* of each player $i \in \mathcal{N}$ is defined as

$$\mathrm{Reg}_i(T) = \max_{\alpha_i \in \mathcal{A}_i} \sum_{t=1}^{T} [u_i(\alpha_i; z_{-i,t}) - u_i(z_t)], \tag{5}$$

and we will say that player $i$ has *no regret* if $\mathrm{Reg}_i(T) = o(T)$.

One of the most widely used policies to achieve no-regret play is the so-called *"follow-the-regularized-leader"* (FTRL) family of algorithms and its variants [45, 46]. To motivate the analysis to come, we begin with an archetypal FTRL method, the *exponential / multiplicative weights* algorithm, also known as HEDGE [4, 5, 11].

**3.1. A gentle start.** We begin our discussion with a "stimulus–response" approach in the spirit of Erev & Roth [19]: First, at each stage $t = 1, 2, \ldots$, every player $i \in \mathcal{N}$ employs a mixed strategy $x_{i,t} \in \mathcal{X}_i$ to select an action $\alpha_{i,t} \in \mathcal{A}_i$. Subsequently, to measure the performance of their pure strategies over time, each player further maintains a score variable which is updated recursively as

$$y_{i\alpha_i,t+1} = y_{i\alpha_i,t} + u_i(\alpha_i; \alpha_{-i,t}) \quad \text{for all } \alpha_i \in \mathcal{A}_i. \tag{6}$$

In words, $y_{i\alpha_i,t}$ simply tracks the cumulative payoff of the pure strategy $\alpha_i \in \mathcal{A}_i$ up to time $t$ (inclusive).[3] As such, this score can be treated as a *propensity* to play a given pure strategy at any given stage: the strategies $\alpha_i \in \mathcal{A}_i$ with the highest propensity scores $y_{i\alpha_i,t+1}$ should be played with higher probability at stage $t + 1$.

The most widely used instantiation of this stimulus-response mechanism is the *logit choice* rule

$$\Lambda_i(y_i) \equiv \frac{(\exp(y_{i\alpha_i}))_{\alpha_i \in \mathcal{A}_i}}{\sum_{\alpha_i \in \mathcal{A}_i} \exp(y_{i\alpha_i})} \tag{7}$$

which means that each player selects an action with probability that is exponentially proportional to its score. In this way, we obtain the *exponential / multiplicative weights* – or HEDGE – algorithm

$$y_{i,t+1} = y_{i,t} + \gamma_t v_i(\alpha_t) \qquad x_{i,t+1} = \Lambda_i(y_{i,t+1}) \qquad \alpha_{i,t+1} \sim x_{i,t+1} \tag{HEDGE}$$

where $\gamma_t$ is the algorithm's "learning rate". For an appetizer to the literature on (HEDGE), see [2, 10, 11, 31, 45] and references therein.

The rest of the methods we discuss below will vary some – or even all – of the components of (HEDGE): the information used to update the players' propensity scores, the way that propensity scores are mapped to mixed strategies, and/or even the way that pure actions are selected. However, all of the methods under study will be characterized by the same "stimulus-response" reinforcement mechanism: actions that seem to be performing better over time are employed with higher probability, up to some "regularization" that incentivizes exploration of underperforming actions.

---

[3]Of course, updating these scores requires the knowledge of the "what if" pure payoffs $u_i(\alpha_i; \alpha_{-i,t})$ at each stage $t$, but we assume for the moment that this information is available (we will relax this assumption later on).

**3.2. The regularized learning template.** In the rest of our paper, we will work with an abstract *regularized learning* (RL) template which builds on the same stimulus-response principle as (HEDGE), while allowing us to simultaneously consider different types of feedback, strategy sampling policies, update structures, etc. To lighten notation below, we will drop the player index $i \in \mathcal{N}$ when the meaning can be inferred from the context; also, to stress the distinction between "strategy-like" and "payoff-like" variables, we will write throughout $\mathcal{Y}_i := \mathbb{R}^{\mathcal{A}_i}$ and $\mathcal{Y} := \prod_i \mathcal{Y}_i$ for the game's "*payoff space*", in direct analogy to $\mathcal{X}_i$ and $\mathcal{X} = \prod_i \mathcal{X}_i$ for the game's *strategy space*.

With all this in hand, consider the following general class of regularized learning methods:

$$\begin{aligned}\text{Aggregate payoff information (stimulus):} \quad & Y_{i,t+1} = Y_{i,t} + \gamma_t \hat{v}_{i,t} \\\text{Update choice probabilities (response):} \quad & X_{i,t+1} = Q_i(Y_{i,t+1})\end{aligned} \tag{RL}$$

In tune with (HEDGE), the various elements of (RL) are defined as follows:

1. $X_{i,t} \in \mathcal{X}_i$ denotes the mixed strategy of player $i$ at time $t = 1, 2, \ldots$
2. $Y_{i,t} \in \mathcal{Y}_i$ is a "score vector" that measures the performance of the player's actions over time.
3. $Q_i \colon \mathcal{Y}_i \to \mathcal{X}_i$ is a "regularized choice map" that maps score vectors to choice probabilities.
4. $\hat{v}_{i,t}$ is a surrogate / approximation of the mixed payoff vector $v_i(X_t)$ of player $i$ at time $t$.
5. $\gamma_t > 0$ is a step-size / sensitivity parameter of the form $\gamma_t \propto 1/t^{\ell_\gamma}$ for some $\ell_\gamma \in [0, 1]$.

In words, at each stage of the process, every player $i \in \mathcal{N}$ observes – or otherwise estimates – a proxy $\hat{v}_{i,t}$ of their individual payoff vector; subsequently, players augment their actions' scores based on this information, they select a mixed strategy via the regularized choice map $Q_i$, and the process repeats. To streamline our presentation, we discuss in detail the precise definition of $\hat{v}$ and $Q$ in Sections 3.3 and 3.4 below, and we present a series of examples of (RL) in Section 3.5 right after.

**3.3. Aggregating payoff information.** As noted above, the main idea of regularized learning is to track the players' payoff vector $v(X_t)$. Importantly, there are several different modeling choices that can be made here: players may have direct access to their payoff vectors (in the full information setting), or some noisy approximation obtained by an inner randomization of the algorithm (e.g., when they receive information on their pure actions); they may have to recreate their payoff vectors altogether (as in the bandit setting), or their estimates may be based on a strategy other than the one they actually played (as in the case of optimistic algorithms).

In all cases, we will represent the surrogate payoff vector $\hat{v}_t$ as

$$\hat{v}_t = v(X_t) + U_t + b_t \tag{8}$$

where $b_t = \mathbb{E}[\hat{v}_t \mid \mathcal{F}_t] - v(X_t)$ and $U_t = \hat{v}_t - \mathbb{E}[\hat{v}_t \mid \mathcal{F}_t]$ respectively denote the offset and the random error of $\hat{v}_t$ relative to $v(X_t)$. To streamline our presentation, we will also assume that $\|b_t\| = \mathcal{O}(1/t^{\ell_b})$ and $\|U_t\| = \mathcal{O}(t^{\ell_\sigma})$ for some $\ell_b, \ell_\sigma \geq 0$; we discuss the specifics of these bounds later in the paper.

**3.4. From scores to strategies.** Regarding the "scores-to-strategies" step of (RL), we will follow the classical approach of Shalev-Shwartz [45] and assume that each player is employing a *regularized choice map* of the general form

$$Q_i(y_i) = \arg\max_{x_i \in \mathcal{X}_i} \{ \langle y_i, x_i \rangle - h_i(x_i) \} \qquad \text{for all } y_i \in \mathcal{Y}_i. \tag{9}$$

In the above, the *regularizer* $h_i \colon \mathcal{X}_i \to \mathbb{R}$ acts as a penalty that smooths out the "hard" argmax correspondence $y_i \mapsto \arg\max_{x_i \in \mathcal{X}_i} \langle y_i, x_i \rangle$. Accordingly, instead of following the "leader" (i.e., playing the strategy with the highest propensity score), players follow the "regularized leader" – that is, they allow for a certain degree of uncertainty in their choice of strategy [10, 33, 45, 46].

To ease notation, we will work with kernelized regularizers of the form $h_i(x_i) = \sum_{\alpha_i \in \mathcal{A}_i} \theta(x_{i\alpha_i})$ for some continuous function $\theta \colon [0, 1] \to \mathbb{R}$ with $\inf_{z \in (0,1]} \theta''(z) > 0$. We will also say that the players' regularizers are *steep* if $\lim_{z \to 0^+} \theta'(z) = -\infty$, and *non-steep* otherwise.

**Example 3.1.** A standard family of kernelized regularizers is given by $\theta(z) = z^\rho / [\rho(\rho - 1)]$ for $\rho \in (0, 1) \cup (1, 2]$ and $\theta(z) = z \log z$ for $\rho = 1$ [10, 31, 33, 53]. This family includes:

• For $\rho = 2$, the quadratic regularizer $\theta(z) = z^2/2$, which yields the Euclidean projection map

$$Q_i(y_i) = \Pi_{\mathcal{X}_i}(y_i) \equiv \arg\min_{x_i \in \mathcal{X}_i} \|y_i - x_i\|_2. \tag{10}$$

- For $\rho = 1$, the *entropic regularizer* $\theta(z) = z \log z$, which yields the logit choice map (7).
- For $\rho = 1/2$, the *fractional power regularizer* $\theta(z) = -4\sqrt{z}$ that underlies the TSALLIS-INF algorithm of [1, 53] (see also Section 3.5 below). ◆

**3.5. Specific algorithms.** We now proceed to discuss some archetypal examples of (RL).

**Algorithm 1** (Follow the regularized leader). The standard *"follow-the-regularized-leader"* (FTRL) method of Shalev-Shwartz & Singer [46] is obtained when players observe their full payoff vectors, that is, $\hat{v}_{i,t} = v_i(X_t)$. In this case, (RL) boils down to the deterministic update rule

$$Y_{i,t+1} = Y_{i,t} + \gamma_t v_i(X_t) \qquad X_{i,t+1} = Q_i(Y_{i,t+1})$$

or, more explicitly

$$X_{i,t+1} = \arg\max_{x_i \in \mathcal{X}_i} \left\{ \sum_{s=1}^{t} \gamma_s u_i(x_i; X_{-i,s}) - h_i(x_i) \right\} \tag{FTRL}$$

For a detailed discussion of (FTRL), see [10, 31, 45]. We only note here that, as a special case, when (FTRL) is run with the logit choice setup of Eq. (7), a standard calculation yields the *exponential / multiplicative weights* (HEDGE) [4, 32, 45, 50]. ◆

**Algorithm 2** (Optimistic FTRL). A notable variant of FTRL – originally due to Popov [40] and subsequently popularized by Rakhlin & Sridharan [41, 42] – is the so-called *optimistic FTRL* method. This scheme employs an "optimistic" correction intended to anticipate future steps, and it updates as

$$Y_{i,t+1} = Y_{i,t} + \gamma_t [2v_i(X_t) - v_i(X_{t-1})] \tag{Opt-FTRL}$$

with $X_{i,t} = Q_i(Y_{i,t})$. As a special case, if (Opt-FTRL) is run with the logit choice map (7), we obtain the familiar update rule known as *optimistic multiplicative weights* (OMW) [15, 41, 42, 47].

Compared to (FTRL), the gain vector $\hat{v}_t = 2v(X_t) - v(X_{t-1})$ of (Opt-FTRL) has offset $b_t = v(X_t) - v(X_{t-1})$ relative to $v(X_t)$. Thus, even though (Opt-FTRL) assumes full access to the players' mixed payoff vectors, it uses this information differently than (FTRL): in particular, the offset of (Opt-FTRL) is non-zero *by design*, not because of some systematic error in the payoff measurement process. ◆

Now, up to this point, we have not detailed how players might observe their full, mixed payoff vectors. This assumption simplifies the analysis immensely, but it is not realistic in applications to e.g., online advertising and network science, where players may only be able to observe their realized payoffs, and have no information about the strategies of other players or actions they did not play. On that account, we describe below a range of *payoff-based* policies where players estimate their counterfactual, "what-if" payoffs *indirectly*.

The most common way to achieve this is via the *importance-weighted estimator*

$$\text{IWE}_{i\alpha_i}(x) = \frac{\mathbb{1}\{\hat{\alpha}_i = \alpha_i\}}{x_{i\alpha_i}} u_i(\hat{\alpha}) \quad \text{for all } \alpha_i \in \mathcal{A}_i, i \in \mathcal{N}, \tag{IWE}$$

where $x \in \mathcal{X}$ is the players' strategy profile, and $\hat{\alpha} \in \mathcal{A}$ is drawn according to $x$. This estimator is at the heart of the online learning literature [10, 11, 31, 45] and it leads to the following methods:

**Algorithm 3** (Bandit FTRL). Plugging (IWE) directly into (RL) yields the *bandit FTRL* policy

$$Y_{i,t+1} = Y_{i,t} + \gamma_t \text{IWE}_i(\hat{X}_t) \qquad X_{i,t+1} = Q_i(Y_{i,t+1}) \tag{B-FTRL}$$

where (IWE) is sampled at the mixed strategy profile

$$\hat{X}_{i,t} = (1 - \delta_t)X_{i,t} + \delta_t \text{ unif}_{\mathcal{A}_i} \tag{11}$$

for some "explicit exploration" parameter $\delta_t \propto 1/t^{\ell_\delta}$, $\ell_\delta > 0$, which specifies the mix between $X_{i,t}$ and the uniform distribution $\text{unif}_{\mathcal{A}_i}$ on $\mathcal{A}_i$. As we discuss in the sequel, this combination of (IWE) with the explicit exploration mechanism (11) means that the surrogate payoff vector $\hat{v}_t = \text{IWE}(\hat{X}_t)$ used to update (B-FTRL) has offset and noise bounded respectively as $b_t = \mathcal{O}(\delta_t)$ and $U_t = \mathcal{O}(1/\delta_t)$.

Two special cases of (B-FTRL) that have attracted significant attention in the literature are:

1. The *exponential weights algorithm for exploration and exploitation* (EXP3) [6, 11, 31], obtained by running (B-FTRL) with the logit choice map (7).

2. The *Tsallis implicitly normalized forecaster* (TSALLIS-INF) [1, 3, 52, 53] that was proposed as a more efficient alternative to EXP3, and which updates as

$$X_{i,t} = \arg\max_{x_i \in \mathcal{X}_i} \left\{ \langle Y_{i,t}, x_i \rangle + 4 \sum_{\alpha_i \in \mathcal{A}_i} \sqrt{x_{i\alpha_i}} \right\} \qquad \text{(TSALLIS-INF)}$$

i.e., as (B-FTRL) with the fractional power regularizer $\theta(z) = -4\sqrt{z}$ of Example 3.1.     ◆

For illustration purposes, we provide some more examples of (RL) in Appendix B.

## 4 First results: resilience to strategic deviations

We are now in a position to begin our analysis of the rationality properties of the players' long-run behavior under (RL). To that end, we should first note that no-regret play may *still* lead to counterintuitive and highly non-rationalizable outcomes, e.g., with all players selecting dominated strategies for all time. The example below is adapted from Viossat & Zapechelnyuk [49].

**Example 4.1.** Consider the $4 \times 4$ symmetric 2-player game with payoff bimatrix

|   | $A$ | $B$ | $C$ | $D$ |
|---|-----|-----|-----|-----|
| $A$ | $(1, 1)$ | $(1, 2/3)$ | $(0, 0)$ | $(0, -1/3)$ |
| $B$ | $(2/3, 1)$ | $(2/3, 2/3)$ | $(-1/3, 0)$ | $(-1/3, -1/3)$ |
| $C$ | $(0, 0)$ | $(0, -1/3)$ | $(1, 1)$ | $(1, 2/3)$ |
| $D$ | $(-1/3, 0)$ | $(-1/3, -1/3)$ | $(2/3, 1)$ | $(2/3, 2/3)$ |

In this game, $B$ and $D$ are strictly dominated for both players by their stronger "twins" ($A$ and $C$ respectively). However, it is easy to check that if both players choose between $(B, B)$ and $(D, D)$ with probability $1/2$ each, the resulting distribution of play $z \in \Delta(\mathcal{A})$ satisfies $u_i(\alpha_i; z_{-i}) - u_i(z) \leq -1/6$ for all $\alpha_i \in \{A, B, C, D\}$, $i = 1, 2$. As a result, the players' regret under $z_t \equiv z$ is *negative*, even though both players play strictly dominated strategies at all times.     ◆

The example above shows unequivocally that

*No-regret play does not suffice to exclude non-rationalizable outcomes.*

In addition, Example 4.1 also shows that predictions based on correlated play are not always appropriate for describing the players' behavior under (RL): the end-state of any regularized learning algorithm will be a closed connected set of mixed strategies, so it is not possible to play *only* $(B, B)$ or $(D, D)$ in the long run. We are thus led to the following natural questions: *What are the rationality properties of long-run play under* (RL)*? Is the players' behavior robust to strategic deviations?*

To study these questions formally, we will focus on the *limit set* $\mathcal{L}(X)$ of $X_t$ under (RL), viz.

$$\mathcal{L}(X) := \bigcap_t \text{cl}\{X_s : s \geq t\} \equiv \{\hat{x} \in \mathcal{X} : X_{t_k} \to \hat{x} \text{ for some subsequence } X_{t_k} \text{ of } X_t\}. \qquad (12)$$

In words, $\mathcal{L}(X)$ is the set of limit points of $X_t$ or, equivalently, the *smallest* subset of $\mathcal{X}$ to which $X_t$ converges. Clearly, the simplest instance of a limit set is when $\mathcal{L}(X)$ is a singleton, i.e., when $X_t$ converges to a point. This case has attracted significant interest in the literature: for example, if $\mathcal{L}(X) = \{x^*\}$ then, for certain special cases of (RL), it is known that $x^*$ is a Nash equilibrium of $\Gamma$ [34]. However, beyond this relatively simple regime, the structure of the limit sets of (RL) could be arbitrarily complicated and their rationality properties are not well-understood.

With this in mind, as a first attempt to study whether the long-run behavior of (RL) is "robust to strategic deviations", we will consider the following notion of *resilience to strategic deviations*:

**Definition 1.** A closed subset $\mathcal{S}$ of $\mathcal{X}$ is *resilient to strategic deviations* – or simply *resilient* – if, for every deviation $x_i \in \mathcal{X}_i$ of every player $i \in \mathcal{N}$, we have

$$u_i(x^*) \geq u_i(x_i, x^*_{-i}) \quad \text{for some } x^* \in \mathcal{S}. \qquad (13)$$

Informally, $\mathcal{S}$ is resilient if every unilateral deviation from $\mathcal{S}$ is deterred by some (possibly different) element thereof. In particular, if $\mathcal{S}$ is a singleton, we immediately recover the definition of a Nash equilibrium; beyond this case however, other examples include the set of undominated strategies of a game, the support face of the equilibria of two-player zero-sum games, etc. Importantly, as we show below, the limit sets of (RL) are resilient *in all games:*

**Theorem 1.** *Let $X_t$, $t = 1, 2, \ldots$, be the sequence of play generated by* (RL) *with step-size / gain parameters $\ell_\gamma > 2\ell_\sigma$ and $\ell_b > 0$. Then, with probability 1, the limit set $\mathcal{L}(X)$ of $X_t$ is resilient.*

*Proof sketch.* The proof of Theorem 1 boils down to two interleaved arguments that we detail in Appendix C. The first hinges on showing that, if $\mathbb{P}(\mathcal{L}(X) = \mathcal{S}) > 0$ for some *non-random* $\mathcal{S} \subseteq \mathcal{X}$, $\mathcal{S}$ must be resilient. This is argued by contradiction: if $p_i \in \mathcal{X}_i$ is a unilateral deviation violating Definition 1, we must also have $\liminf_{t \to \infty}[u_i(p_i; X_{-i,t}) - u_i(X_t)] > 0$ with positive probability. However, the existence of a strategy that consistently outperforms $X_t$ runs contrary to the fact that strategies that (RL) selects against underperforming strategies. We make this intuition precise via an energy argument that leverages a series of results from martingale limit theory (which is where the requirements for $\gamma_t$, $b_t$ and $U_t$ come in). Then, to get the stronger statement that the *random* set $\mathcal{L}(X)$ is resilient w.p.1, we show that the above remains true if $p_i$ is replaced by a deviation $q_i$ which is close enough to $p_i$ and has *rational* entries. Since there is a countable number of such profiles, we can use a union bound on an enumeration of the rationals to isolate a deviation witnessing the negation of Definition 1; our claim then follows by applying our argument for non-random sets. ∎

Theorem 1 is our first universal guarantee for (RL), so some remarks are in order. First, we should point out that the requirements $\ell_b > 0$ and $2\ell_\sigma < \ell_\gamma$ are a priori *implicit* because they depend on the offset and magnitude statistics of the feedback sequence $\hat{v}_t$. However, in most learning algorithms, these quantities are under the *explicit* control of the players: for example, as we show in Appendix B, Algorithm 2 has $\ell_b = \ell_\gamma$ while, for Algorithm 3, we have $\ell_b = \ell_\sigma = \ell_\delta$. In this way, when instantiated to Algorithms 1–3 (and special cases thereof), Theorem 1 yields the following corollary:

**Corollary 1.** *Suppose that Algorithms 1–3 are run with $\ell_\gamma \in (0, 1]$ and, for Algorithm 3, $\ell_\delta \in (0, \ell_\gamma/2)$. Then, with probability 1, the limit set $\mathcal{L}(X)$ of $X_t$ is resilient.*

Now, since Theorem 1 applies to all games, it would seem to provide a universally positive answer to whether (RL) is robsut to strategic deviations. However, this is not so: a direct calculation shows that the face of $\mathcal{X}$ that is spanned by the dominated strategies $(B, B)$ and $(D, D)$ of Example 4.1 *is* resilient, so Theorem 1 cannot exclude convergence to a set where dominated strategies survive. Thus, just like no-regret play, the notion of resilience does not suffice by itself to capture the idea of rational behavior. This is because, albeit natural, resilience is too lax to provide a meaningful link between robustness to unilateral deviations – a *game-theoretic* requirement – and stability under regularized learning – a *dynamic* requirement. We address this question in detail in the next section.

## 5 A characterization of strategic stability under regularized learning

Similar to the set of pure strategies that arise from no-regret play, the main limitation of resilience is that a payoff-improving deviation may be countered by an action profile where the deviator also switched to a *different* strategy; in other words, resilience is not a *self-enforcing* barrier to deviations. In view of this, we will focus below on a much more stringent criterion of strategic stability, namely that *any* deviation from the set in question incurs a cost to the deviating agent.

**Club sets.** To make all this precise, define the *better-reply correspondence* of player $i \in \mathcal{N}$ as
$$\mathtt{btr}_i(x) = \{x_i' \in \mathcal{X}_i : u_i(x_i'; x_{-i}) \geq u_i(x)\} \tag{14}$$
and write $\mathtt{btr} = \prod_i \mathtt{btr}_i$ for the product correspondence $\mathtt{btr}(x) = \mathtt{btr}_1(x) \times \cdots \times \mathtt{btr}_N(x)$. [In words, $\mathtt{btr}_i$ assigns to each $x \in \mathcal{X}$ those strategies of player $i$ that are (weakly) better against $x$ than $x_i$.] In addition, given a product of pure strategies $\mathcal{C} = \prod_{i \in \mathcal{N}} \mathcal{C}_i$ with $\mathcal{C}_i \subseteq \mathcal{A}_i$ for all $i \in \mathcal{N}$, let $\mathcal{S} = \Delta(\mathcal{C})$ denote the span of $\mathcal{C}$, and let $\mathcal{P}(\mathcal{X})$ denote the collection of all such sets. We then say that $\mathcal{S} \in \mathcal{P}(\mathcal{X})$ is *closed under better replies* – a *club set* for short – if it is closed under $\mathtt{btr}$, i.e., $\mathtt{btr}(\mathcal{S}) \subseteq \mathcal{S}$; finally, $\mathcal{S}$ is said to be *minimally club* (m-club) if it does not admit a proper club subset.

Of course, the entire strategy space $\mathcal{X}$ is closed under better replies so, a priori, club sets could also contain dominated strategies and / or other non-rationalizable outcomes. By contrast, *minimal* club sets are much more rigid in their relation to rational behavior because any unilateral deviation from an m-club set is *costly*, and m-club sets are *minimal* in this regard. On that account, m-club sets can be seen as *the closest setwise analogue to strict Nash equilibria*.

This analogy is accentuated further by the following properties of m-club sets, all due to Ritzberger & Weibull [43], who introduced the concept:

1. Every game admits an m-club set; and if this set is a singleton, then it is a *strict* Nash equilibrium.
2. Any m-club set $\mathcal{S}$ is *fixed* under better replies, that is, $\mathtt{btr}(\mathcal{S}) = \mathcal{S}$ (implying in turn that $\mathcal{S}$ cannot contain any dominated strategies, including iteratively dominated ones).
3. Any m-club set $\mathcal{S}$ contains an *essential equilibrium component*, i.e., a component of Nash equilibria such that every small perturbation of the game admits a nearby equilibrium; in addition, this component has *full support* on $\mathcal{S}$, i.e., it employs all pure strategy profiles that lie in $\mathcal{S}$.[4]

Going back to our online learning setting, the above leads to the following natural set of questions:

> *Are club sets (minimal or not) stable under the dynamics of regularized learning?*
> *Are they attracting? And, if so, are they the only such sets?*

Any answer to these questions – positive or negative – would be an important step in delineating the relation between *strategic stability* (in the above sense) and *dynamic stability* under (RL). To that end, we start by formalizing some notions of dynamic stability that will be central in the sequel:

**Definition 2.** Fix some subset $\mathcal{S}$ of $\mathcal{X}$ and a tolerance level $\epsilon > 0$. We then say that $\mathcal{S}$ is:

1. *Stochastically stable* if, for every neighborhood $\mathcal{U}$ of $\mathcal{S}$ in $\mathcal{X}$, there exists a neighborhood $\mathcal{U}_1$ of $\mathcal{S}$ such that
$$\mathbb{P}(X_t \in \mathcal{U} \text{ for all } t = 1, 2, \dots) \geq 1 - \epsilon \quad \text{whenever } X_1 \in \mathcal{U}_1. \tag{15}$$
2. *Stochastically attracting* if there exists a neighborhood $\mathcal{U}_1$ of $\mathcal{S}$ such that
$$\mathbb{P}(\lim_{t\to\infty} \mathrm{dist}(X_t, \mathcal{S}) = 0) \geq 1 - \epsilon \quad \text{whenever } X_1 \in \mathcal{U}_1. \tag{16}$$
3. *Stochastically asymptotically stable* if it is stochastically stable and attracting.
4. *Irreducibly stable* if $\mathcal{S}$ is stochastically asymptotically stable and it does not admit a strictly smaller stochastically asymptotically subset $\mathcal{S}'$ with $\mathrm{supp}(\mathcal{S}') \subsetneq \mathrm{supp}(\mathcal{S})$.

With all this in hand, our main result below provides a sharp characterization of strategic stability in the context of regularized learning:

**Theorem 2.** *Fix some set $\mathcal{S} \in \mathcal{P}(\mathcal{X})$ and suppose that (RL) is run with a steep regularizer and step-size / gain parameters $\ell_\gamma \in [0, 1]$, $\ell_b > 0$, and $\ell_\sigma < 1/2$. Then:*

1. $\mathcal{S}$ *is stochastically asymptotically stable under (RL) if and only if it is a club set.*
2. $\mathcal{S}$ *is irreducibly stable under (RL) if and only if it is an m-club set.*

In addition, we also get the following convergence rate estimates for club sets:

**Theorem 3.** *Let $\mathcal{S} \in \mathcal{P}(\mathcal{X})$ be a club set, and let $X_t$, $t = 1, 2, \dots$, be the sequence of play generated by (RL) with parameters $\ell_\gamma \in [0, 1]$, $\ell_b > 0$, and $\ell_\sigma < 1/2$. Then, for all $\epsilon > 0$, there exists an (open, unbounded) initialization domain $\mathcal{D} \subseteq \mathcal{Y}$ such that, with probability at least $1 - \epsilon$, we have*
$$\mathrm{dist}(X_t, \mathcal{S}) \leq C\varphi\left(c_1 - c_2 \sum_{s=1}^{t} \gamma_s\right) \quad \text{whenever } Y_1 \in \mathcal{D} \tag{17}$$
*where $C, c_1, c_2$ are constants ($C, c_2 > 0$), and the rate function $\varphi$ is given by $\varphi(z) = (\theta')^{-1}(z)$ if $z > \lim_{z\to 0^+} \theta'(z)$, and $\varphi(z) = 0$ otherwise.*

Specifically, if we instantiate Theorem 3 to Algorithms 1–3, we get the explicit estimates:

**Corollary 2.** *Suppose that Algorithms 1–3 are run with $\ell_\gamma \in [0, 1]$ and, for Algorithm 3, $\ell_\delta \in (0, 1/2)$. Then, with notation as in Theorem 3, $X_t$ converges to $\mathcal{S}$ at a rate of*
$$\mathrm{dist}(X_t, \mathcal{S}) \leq C \cdot \begin{cases} [1 - c\sum_{s=1}^{t}\gamma_s]_+ & \text{if } \theta(z) = z^2/2 & \text{\# quadratic regularization} \\ \exp\left(-c\sum_{s=1}^{t}\gamma_s\right) & \text{if } \theta(z) = z\log z & \text{\# entropic regularization} \\ 1/\left(c + \sum_{s=1}^{t}\gamma_s\right)^2 & \text{if } \theta(z) = -4\sqrt{z} & \text{\# fractional regularization} \end{cases} \tag{18}$$
*for positive constants $C, c > 0$. In particular, the projection-based variants of Algorithms 1–3 converge to m-club sets in a **finite** number of steps.*

---

[4] Formally, a component $\mathcal{X}^*$ of Nash equilibria of $\Gamma$ is *essential* if, for all $\varepsilon > 0$, there exists $\delta > 0$ such that any perturbation of the payoffs of $\Gamma$ by at most $\delta$ produces a Nash equilibrium that is $\varepsilon$-close to $\mathcal{X}^*$ [48]. This property – known as "*essentiality*" – has a long history as one of the strictest setwise solution refinements in game theory; in particular, it satisfies all the seminal *strategic stability* requirements of Kohlberg & Mertens [30], including robustness to strategic payoff perturbations. For an in-depth discussion, see van Damme [48].

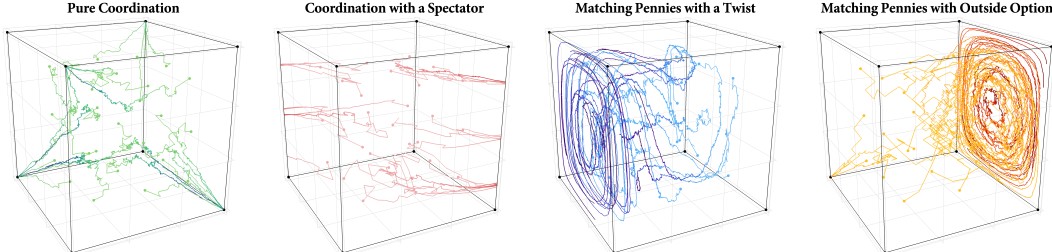

**Figure 1:** The long-run behavior of EXP3 (Algorithm 3) in four representative $2 \times 2 \times 2$ games. In all cases, the dynamics converge to m-club sets, either *strict equilibria* themselves, or spanning an *essential component* of Nash equilibria. The details of the numerics and the games being played are provided in the appendix.

*Proof sketch.* The proof of Theorems 2 and 3 is quite involved so we defer it to Appendix D. At a high level, it hinges on constructing a family of "primal-dual" energy functions, one per pure deviation from the set $\mathcal{S}$ under study. If unilateral deviations from $\mathcal{S}$ incur a cost to the deviator (that is, if $\mathcal{S}$ is club), these energy functions can be "bundled together" to produce a suitable Lyapunov-like function for $\mathcal{S}$. In more detail, the minimization of each individual energy function implies that the score variable $Y_t$ of (RL) diverges along an "astral direction" in the payoff space $\mathcal{Y}$ – i.e., it escapes to infinity along the interior of a certain convex cone of $\mathcal{Y}$ [18]. Because this minimization occurs at infinity, the aggregation of offsets and random errors in (RL) affords some extra "wiggle room" in our martingale analysis, so we are able to show that $X_t = Q(Y_t)$ remains close to $\mathcal{S}$ under a much wider range of parameters compared to Theorem 1. Then, a series of convex analysis arguments in the spirit of [33] coupled with the definition of $Q$ allows us to show that the escape of $Y_t$ along the intersection of all these cones implies convergence to $\mathcal{S}$ at the specified rate.

On the converse side, if an asymptotically stable set is not club, we can find a non-costly (and possibly profitable) deviation $z$ from $\mathcal{S}$ which is selected against by (RL). However, this extinction runs contrary to the reinforcement of better replies under (RL), an argument which can be made precise by applying the martingale law of large numbers to $\langle Y_t, z \rangle$ [25]. The irreducible stability of m-club sets then follows by invoking this criterion reductively for any potentially stable subset $\mathcal{S}'$ of $\mathcal{S}$. ∎

**Discussion and remarks.** Theorems 2 and 3 are our main results linking dynamic and strategic stability, so we conclude with a series of remarks. First, we should note that Theorem 2 can be summed up as follows: *a product of pure strategies is (minimally) closed under better replies if and only if its span is (irreducibly) stable under regularized learning.* Importantly, this equivalence is based solely on the game's payoff data: it does not depend on the specific choices underlying (RL), including the choice map employed by each player, whether some players are using an optimistic adjustment or not, if they have access to their full payoff vectors, etc. As such, this equivalence provides a crisp operational criterion for identifying which pure strategy combinations ultimately persist under regularized learning – and, via Theorem 3, *how fast* this identification takes place.

In this light, Theorem 2 essentially states that the only robust prediction that can be made for the outcome of a regularized learning process is (minimal) closedness under better replies. This interpretation has significant cutting power for the emergence of rational behavior. To begin, in terms of equilibrium play, it effortlessly implies that a pure strategy profile is stochastically asymptotically stable under (RL) if and only if it is a strict Nash equilibrium. A version of this equivalence was only recently proved in [20] and [22] (in continuous and discrete time respectively), so Theorem 2 can be seen as a far-reaching generalization of these recent results. More to the point, since every m-club set $\mathcal{S}$ contains an essential equilibrium component that is fully supported in $\mathcal{S}$, Theorem 2 also provides an important link between dynamic and structural stability: if an equilibrium – or a component of equilibria – is not robust to perturbations of the underlying game, *it cannot be robustly identified by a regularized learning process* (and vice versa). This remark is of particular importance for extensive-form games as such games often have non-generic equilibrium components that cannot be treated otherwise by the existing theory.

Finally, we should stress that Theorems 2 and 3 guarantee convergence even with a constant step-size. Together with the finite-time convergence guarantees of Corollary 2 for projection-based methods, this feature is a testament to the robustness of club sets as, in the presence of uncertainty, convergence almost always requires a vanishing step-size which can slow convergence down to a crawl. We find this robust convergence landscape particularly intriguing for future research on the topic.

## Acknowledgments and Disclosure of Funding

PM is also with the Archimedes Research Unit – Athena RC – Department of Mathematics, National & Kapodistrian University of Athens. This work has been partially supported by the French National Research Agency (ANR) in the framework of the "Investissements d'avenir" program (ANR-15-IDEX-02), the LabEx PERSYVAL (ANR-11-LABX-0025-01), MIAI@Grenoble Alpes (ANR-19-P3IA-0003), and project MIS 5154714 of the National Recovery and Resilience Plan Greece 2.0 funded by the European Union under the NextGenerationEU Program

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

# A  Auxiliary results

In this appendix we collect some basic properties of the regularized choice maps and some results from probability theory that will be useful in the sequel.

**A.1.  Regularized choice maps and their properties.**  Thoughout this appendix, we will suppress the player index $i \in \mathcal{N}$, and we will follow standard conventions in convex analysis [44] that treat $h$ as an extended-real-valued function $h \colon \mathcal{V} \to \mathbb{R} \cup \{\infty\}$ with $h(x) = \infty$ for all $x \in \mathcal{V} \setminus \mathcal{X}$. With this in mind, the subdifferential of a $h$ at $x \in \mathcal{X}$ is defined as

$$\partial h(x) \coloneqq \{y \in \mathcal{Y} : h(x') \geq h(x) + \langle y, x' - x \rangle \text{ for all } x' \in \mathcal{X}\}, \tag{A.1}$$

where $\mathcal{Y}$ denotes here the algebraic dual $\mathcal{V}^*$ of $\mathcal{V}$. Accordingly, the *domain of subdifferentiability* of $h$ is $\operatorname{dom} \partial h \coloneqq \{x \in \operatorname{dom} h : \partial h \neq \varnothing\}$, and the convex conjugate of $h$ is defined as

$$h^*(y) = \max_{x \in \mathcal{X}} \{\langle y, x \rangle - h(x)\} \tag{A.2}$$

for all $y \in \mathcal{Y}$. We then have the following basic results.

**Lemma A.1.** *Let $h$ be a regularizer on $\mathcal{X}$, and let $Q \colon \mathcal{Y} \to \mathcal{X}$ be the induced choice map. Then:*

1. *$Q$ is single-valued, and, for all $x \in \mathcal{X}$, $y \in \mathcal{Y}$, we have $x = Q(y) \iff y \in \partial h(x)$.*
2. *For all $x \in \operatorname{ri} \mathcal{X}$, we have $\partial h(x) = \{(\theta'(x_\alpha) + \mu)_{\alpha \in \mathcal{A}} : \mu \in \mathbb{R}\}$.*
3. *For all $y \in \mathcal{Y}$, we have $Q(y) = \nabla h^*(y)$.*
4. *$Q$ is $(1/K)$-Lipschitz continuous with $K \coloneqq \inf_{(0,1]} \theta''(z)$. In particular, as a special case, the logit choice map $\Lambda$ is $1$-Lipschitz continuous in the $(L^1, L^\infty)$ pair of norms on $\mathcal{Y}$ and $\mathcal{X}$ respectively.*
5. *If $y_\alpha - y_{\alpha'} \to -\infty$ for some $\alpha' \neq \alpha$, then $Q_\alpha(y) \to 0$.*

*Remark.* Some of the properties presented in Lemma A.1 are well known in the literature on regularized learning methods (see e.g., [33] and references therein), but we provide a proof of the entire lemma for completeness. ♦

*Proof of Lemma A.1.* For the first property of $Q$, note that the maximum in (9) is attained for all $y \in \mathcal{Y}$ because $h$ is lower-semicontinuous (l.s.c.) and strongly convex. Furthermore, $x$ solves (9) if and only if $y - \partial h(x) \ni 0$, i.e., if and only if $y \in \partial h(x)$.

For our second claim, if $x \in \operatorname{ri}(\mathcal{X})$, the first-order stationarity conditions for the convex problem (9) that defines $Q$ become

$$y_\alpha - \theta'(x_\alpha) = \mu \quad \text{for all } \alpha \in \mathcal{A}, \tag{A.3}$$

because the inequality constraints $x_\alpha \geq 0$ are all inactive (recall that $x \in \operatorname{ri}(\mathcal{X})$ by assumption). Now, by the first part of the theorem we have $x = Q(y)$ if and only if $y \in \partial h(x)$, so we conclude that $\partial h(x) = \{(\theta'(x_\alpha) + \mu)_{\alpha \in \mathcal{A}} : \mu \in \mathbb{R}\}$, as claimed.

For the fourth item, the expression $Q = \nabla h^*$ is an immediate consequence of Danskin's theorem, while the Lipschitz continuity of $Q$ follows from standard results, see e.g., [44, Theorem 12.60(b)].

For our last claim, let $y_t$ be a sequence in $\mathcal{Y}$ such that $y_{\alpha,t} - y_{\alpha',t} \to -\infty$ and let $x_t = Q(y_t)$. Then, by descending to a subsequence if necessary, assume there exists some $\varepsilon > 0$ such that $x_{\alpha,t} \geq \varepsilon > 0$ for all $t$. Then, by the defining relation $Q(y) = \arg\max\{\langle y, x \rangle - h(x)\}$ of $Q$, we have:

$$\langle y_t, x_t \rangle - h(x_t) \geq \langle y_t, x' \rangle - h(x') \tag{A.4}$$

for all $x' \in \mathcal{X}$. Therefore, taking $x'_t = x_t + \varepsilon(e_{\alpha'} - e_\alpha)$, we readily obtain

$$\varepsilon(y_{\alpha,t} - y_{\alpha',t}) \geq h(x_t) - h(x'_t) \geq \min h - \max h \tag{A.5}$$

which contradicts our original assumption that $y_{\alpha,t} - y_{\alpha',t} \to -\infty$. With $\mathcal{X}$ compact, the above shows that $x^*_\alpha = 0$ for any limit point $x^*$ of $x_t$, i.e. $Q_\alpha(y_t) \to 0$. ∎

The second collection of results concerns the *Fenchel coupling*, an energy function that was first introduced in [33, 34] and is defined as follows:

$$F(p, y) = h(p) + h^*(y) - \langle y, p \rangle \quad \text{for all } p \in \mathcal{X} \text{ and } y \in \mathcal{Y}. \tag{A.6}$$

This coupling will play a major role in the proofs of Theorem 1, so we prove two of its most basic properties below.

**Lemma A.2.** *For all $p \in \mathcal{X}$ and all $y, y' \in \mathcal{Y}$, we have:*

a) $\quad F(p, y) \geq \frac{1}{2} K \|Q(y) - p\|^2.$ $\qquad\qquad\qquad\qquad\qquad\qquad\qquad\qquad$ (A.7a)

b) $\quad F(p, y') \leq F(p, y) + \langle y' - y, Q(y) - p \rangle + \frac{1}{2K} \|y' - y\|_\infty^2.$ $\qquad\qquad$ (A.7b)

*In particular, if $h(0) = 0$, we have*

$$(K/2)\|Q(y)\|^2 \leq h^*(y) \leq -\min h + \langle y, Q(y) \rangle + (2/K)\|y\|_\infty^2 \quad \text{for all } y \in \mathcal{Y}. \qquad \text{(A.8)}$$

*Proof of Lemma A.2.* By the strong convexity of $h$ relative to $\|\cdot\|$ (cf. Lemma A.1), we have

$$
\begin{aligned}
h(x) + t\langle y, p - x \rangle &\leq h(x + t(p - x)) \\
&\leq t h(p) + (1 - t)h(x) - \tfrac{1}{2} K t(1 - t)\|x - p\|^2,
\end{aligned}
\qquad \text{(A.9)}
$$

leading to the bound

$$\tfrac{1}{2} K(1 - t)\|x - p\|^2 \leq h(p) - h(x) - \langle y, p - x \rangle = F(p, y) \qquad\qquad \text{(A.10)}$$

for all $t \in (0, 1]$. The bound (A.7a) then follows by letting $t \to 0^+$ in (A.10).

For our second claim, we have

$$
\begin{aligned}
F(p, y') &= h(p) + h^*(y') - \langle y', p \rangle \\
&\leq h(p) + h^*(y) + \langle y' - y, \nabla h^*(y) \rangle + \frac{1}{2K} \|y' - y\|_\infty^2 - \langle y', p \rangle \\
&= F(p, y) + \langle y' - y, Q(y) - p \rangle + \frac{1}{2K} \|y' - y\|_\infty^2,
\end{aligned}
\qquad \text{(A.11)}
$$

where the inequality in the second line follows from the fact that $h^*$ is $(1/K)$-strongly smooth [44, Theorem 12.60(e)]. $\qquad\qquad\qquad\qquad\qquad\qquad\qquad\qquad\qquad\qquad\qquad\qquad\blacksquare$

**A.2. Basic results from probability theory.** We conclude this appendix with some useful results from probability theory that we will use freely throughout the sequel. For a complete treatment, we refer the reader to Hall & Heyde [25].

**Lemma A.3** (Azuma-Hoeffding inequality). *Let $M_t \in \mathbb{R}$, $t = 1, 2, \ldots$, be a martingale with $\|M_t - M_{t-1}\|_\infty \leq \sigma_t$ (a.s.). Then, for all $\eta > 0$, we have*

$$\mathbb{P}\left(|M_t| \leq \left(2\log(2t^2/\eta) \sum_{s=1}^t \sigma_s^2\right)^{1/2} \text{ for all } t\right) \geq 1 - \eta. \qquad \text{(A.12)}$$

**Lemma A.4** (Kolmogorov's inequality). *Let $Z_t \in \mathbb{R}$, $t = 1, 2, \ldots$, be a martingale difference sequence that is bounded in $L^2$. Then:*

$$\mathbb{P}\left(\max_{s \leq t} \sum_{\ell=1}^s Z_\ell \geq \varepsilon\right) \leq \frac{1}{\varepsilon^2} \mathbb{E}\left[\left(\sum_{s=1}^t Z_s\right)^2\right] \quad \text{for all } \varepsilon > 0. \qquad \text{(A.13)}$$

**Lemma A.5** (Doob's maximal inequality). *Let $Z_t \in \mathbb{R}$, $t = 1, 2, \ldots$, be a martingale difference sequence that is bounded in $L^p$ for some $p \geq 1$. Then*

$$\mathbb{P}\left(\max_{s \leq t} |Z_s| > \varepsilon\right) \leq \frac{1}{\varepsilon^p} \mathbb{E}\left[|Z_t|^p\right] \quad \text{for all } \varepsilon > 0. \qquad \text{(A.14)}$$

**Lemma A.6** (Burkholder–Davis–Gundy inequality). *Let $Z_t$, $t = 1, 2, \ldots$, be a martingale difference sequence in $\mathbb{R}^n$. Then, for all $p > 1$, there exist constants $c_p, C_p$ that depend only on $p$ and are such that*

$$c_p \, \mathbb{E}\left[\sum_{s=1}^t \|Z_s\|_2^2\right]^{p/2} \leq \mathbb{E}\left[\max_{s \leq t} \left\|\sum_{\ell=1}^s Z_\ell\right\|_2^p\right] \leq C_p \, \mathbb{E}\left[\sum_{s=1}^t \|Z_s\|_2^2\right]^{p/2}. \qquad \text{(A.15)}$$

**Lemma A.7** (Robbins–Siegmund). *Let $\mathcal{F}_t$, $t = 1, 2, \ldots$, be a filtration on a complete probability space $(\Omega, \mathcal{F}, \mathbb{P})$, and suppose that the sequences $X_t$, $L_t$ and $K_t$ $\mathcal{F}_t$-measurable, nonnegative, and such that*

$$\mathbb{E}[X_{t+1} \mid \mathcal{F}_t] \leq X_t(1 + L_t) + K_t \quad \text{with probability 1.} \qquad \text{(A.16)}$$

*Then, $X_t$ converges to some random variable $X_\infty$ with probability 1 on the event $\left\{\sum_{t=1}^\infty L_t < \infty \text{ and } \sum_{t=1}^\infty K_t < \infty\right\}$.*

# B Specific algorithms and their properties

**B.1. Known algorithms as special cases of** (RL)**.** To complement our analysis in the main part of our paper, we detail below how Algorithms 1–3 can be recast in the general framework of (RL). To lighten notation, we will assume that $b_t$, $U_t$ and $\hat{v}_t$ are respectively bounded as

$$\|b_t\|_\infty \le B_t \qquad \|U_t\|_\infty \le \sigma_t \qquad \text{and} \qquad \|\hat{v}_t\|_\infty \le M_t \tag{B.1}$$

and we will set

$$G := \max_{i \in \mathcal{N}} \max_{\alpha \in \mathcal{A}} |v_i(\alpha)| \tag{B.2}$$

so we can take $M_t = G + B_t + \sigma_t$ in (B.1). We will also make free use of the fact that $v$ is Lipschitz continuous on $\mathcal{X}$, and we will write $L$ for its Lipschitz modulus in the $(L^1, L^\infty)$ pair of norms on $\mathcal{X}$ and $\mathcal{Y}$ respectively, viz.

$$\|v(x') - v(x)\|_\infty \le L\|x' - x\|_1 \qquad \text{for all } x, x' \in \mathcal{X}. \tag{B.3}$$

We now proceed to establish the required bounds for Algorithms 1–3:

**Algorithm 1.** Since $\hat{v}_t = v(X_t)$, we readily get $b_t = U_t = 0$ by definition, so Algorithm 1 fits the scheme (RL) for free with $\ell_b = \infty$, $\ell_\sigma = 0$. ♦

**Algorithm 2.** For the case of (Opt-FTRL), we have $\hat{v}_t = 2v(X_t) - v(X_{t-1})$ so $b_t = v(X_t) - v(X_{t-1})$, which is $\mathcal{F}_t$-measurable. We thus get

$$
\begin{aligned}
\|b_t\|_\infty = \|\mathbb{E}[\hat{v}_t \mid \mathcal{F}_t] - v(X_t)\|_\infty &\le \mathbb{E}[\|v(X_t) - v(X_{t-1})\|_\infty \mid \mathcal{F}_t] \\
&\le L\,\mathbb{E}[\|X_t - X_{t-1}\| \mid \mathcal{F}_t] && \text{\# by (B.3)} \\
&= L\,\mathbb{E}[\|Q(Y_t) - Q(Y_{t-1})\|_\infty \mid \mathcal{F}_t] && \text{\# by (Opt-FTRL)} \\
&\le (L/K)\,\mathbb{E}[\|Y_t - Y_{t-1}\|_\infty \mid \mathcal{F}_t] && \text{\# by Lemma A.1} \\
&\le \gamma_t (L/K)\,\mathbb{E}[2v(X_t) - v(X_{t-1}) \mid \mathcal{F}_t] && \text{\# by (Opt-FTRL)} \\
&\le 3LG/K \cdot \gamma_t && \text{\# by (B.2)} \\
&= \mathcal{O}(\gamma_t) = \mathcal{O}(1/t^{\ell_\gamma}) && \text{(B.4)}
\end{aligned}
$$

Moreover, given that $\hat{v}$ is $\mathcal{F}_t$-measurable, we readily get $U_t = 0$. ♦

**Algorithm 3.** Since $\hat{\alpha}_t$ is sampled according to $\hat{X}_t = (1 - \delta_t)X_{i,t} + \delta_t \, \mathrm{unif}_{\mathcal{A}_i}$ (cf. Eq. (11) in Section 3), we readily obtain $\mathbb{E}[\hat{v}_{i,t} \mid \mathcal{F}_t] = v_i(\hat{X}_t)$, and hence, by (B.3), we get

$$B_t = \mathcal{O}(\|\hat{X}_t - X_t\|) = \mathcal{O}(\delta_t) = \mathcal{O}(1/t^{\ell_\delta}). \tag{B.5}$$

Moreover, since $\hat{X}_{i\alpha_i,t} \ge \delta_t/A_i$, it follows that $\|\hat{v}_t\|_\infty = \mathcal{O}(1/\delta_t) = \mathcal{O}(t^{\ell_\delta})$. ♦

For comparison purposes, we illustrate the algorithms' behavior in a simple $2 \times 2 \times 2$ game in Fig. 2 in Appendix E.

**B.2. Further algorithms and illustrations.** To demonstrate the breadth of (RL) as an algorithmic template, we provide below some more examples of algorithms from the game-theoretic literature that can be recast as special cases thereof (see also Table 1 for a recap).

**Algorithm 4** (Mirror-prox)**.** A progenitor of (Opt-FTRL) is the so-called *mirror-prox* (MP) algorithm [29, 37], which updates as:

$$
\begin{aligned}
\tilde{Y}_t &= Y_t + \gamma_t v(X_t) & Y_{t+1} &= Y_t + \gamma_t v(\tilde{X}_t) \\
\tilde{X}_t &= Q(\tilde{Y}_t) & X_{t+1} &= Q(Y_{t+1}).
\end{aligned} \tag{MP}
$$

The main difference between (MP) and (Opt-FTRL) is that the former utilizes two surrogate gain vectors per iteration – meaning in particular that the interim, leading state $\tilde{X}_t$ is generated with payoff information from $X_t$, not $\tilde{X}_{t-1}$. This method has been used extensively in the literature for solving variational inequalities and two-player, zero-sum games, cf. Juditsky et al. [29] and references therein.

A calculation similar to that for (Opt-FTRL) shows that Algorithm 4 has $B_t = \mathcal{O}(1/t^{\ell_\gamma})$ and $\sigma_t = 0$ because the algorithm has no further randomization. ♦

| | Representative | Regularizer ($\theta$) | Feedback | Bias ($B_t$) | Variance ($\sigma_t$) |
|---|---|---|---|---|---|
| Algorithm 1 | HEDGE | $z \log z$ | full info | $0$ | $0$ |
| Algorithm 2 | OMW | $z \log z$ | full info | $\mathcal{O}(1/t^{\ell_\gamma})$ | $0$ |
| Algorithm 3 | EXP3 | $z \log z$ | payoff | $\mathcal{O}(1/t^{\ell_\delta})$ | $\mathcal{O}(t^{\ell_\delta})$ |
| Algorithm 3 | TSALLIS-INF | $-4\sqrt{z}$ | payoff | $\mathcal{O}(1/t^{\ell_\delta})$ | $\mathcal{O}(t^{\ell_\delta})$ |
| Algorithm 4 | MP | general | full info | $\mathcal{O}(1/t^{\ell_\gamma})$ | $0$ |
| Algorithm 5 | CMW | $z \log z$ | full info | $\mathcal{O}(1/t^{\ell_\delta})$ | $0$ |

**Table 1:** A range of algorithms adhering to the general template (RL) and their bias and variance characteristics when run with a step-size sequence of the form $\gamma_t = \gamma/t^{\ell_\gamma}$, $\ell_\gamma \in (0, 1]$, and, where applicable, a sampling parameter $\delta_t = \delta/t^{\ell_\delta}$.

**Algorithm 5** (Clairvoyant multiplicative weights). A recent variant of the exponential / multiplicative weights (HEDGE) algorithm is the so-called *clairvoyant multiplicative weights* (CMW) algorithm [39]

$$Y_{i,t+1} = Y_{i,t} + \gamma_t v_i(X_{t+1}) \qquad X_{i,t+1} = \Lambda_i(Y_{i,t+1}). \tag{CMW}$$

The main difference between (CMW) and (HEDGE) is that the proxy payoff vector $\hat{v}_t$ in (CMW) is based on the *future* state $X_{t+1}$ and *not* the current state $X_t$. To perform this "clairvoyant" update, the players of the game must coordinate to solve an implicit fixed point problem, so (CMW) is only meaningful when one has access to the payoff function $v(\cdot)$. In this regard, (CMW) can be seen as a Bregman proximal point method in the general spirit of Bauschke et al. [8].

To cast (CMW) as an instance of the generalized template (RL), simply note that the sequence of input signals is given by $\hat{v}_t = v(X_{t+1})$, so $U_t = 0$ and $b_t = v(X_{t+1}) - v(X_t) = \mathcal{O}(\gamma_t) = \mathcal{O}(1/t^{\ell_\gamma})$. ♦

## C Proof of Theorem 1

Our main goal in this appendix will be to prove Theorem 1 on the resilience properties of (RL). For convenience, we restate below the relevant result for ease of reference:

**Theorem 1.** *Let $X_t$, $t = 1, 2, \ldots$, be the sequence of play generated by* (RL) *with step-size / gain parameters $\ell_\gamma > 2\ell_\sigma$ and $\ell_b > 0$. Then, with probability $1$, the limit set $\mathcal{L}(X)$ of $X_t$ is resilient.*

*Proof.* Our proof that $\mathcal{L}(X)$ is resilient hinges on an energy-based technique that we will employ repeatedly in other parts of our analysis. To begin, introduce a player-strategy deviation pair $(i, z_i)$, and say that a set is resilient *to* $(i, z_i)$ if there exists an element of the set, say $x^*$, which counters said deviation, i.e., such that $u_i(x^*) \geq u_i(z_i; x^*_{-i})$. In this specific case, our proof proceeds by contradiction, namely by assuming that, with positive probability, $\mathcal{L}(X)$ is *not* resilient to $(i, z_i)$. The main steps of our proof unfold as follows:

**Step 1.** *Assume that $\mathcal{L}(X)$ is not resilient to $(i, z_i)$ with positive probability. Then there exists $c, \epsilon, t_0 > 0$ such that*

$$\mathbb{P}\big(u_i(z_i; X_{t,-i}) \geq u_i(X_t) + c \text{ for all } t \geq t_0\big) \geq \epsilon. \tag{C.1}$$

*Proof of Step 1.* The function $f : x \in \mathcal{X} \mapsto u_i(z_i; x_{-i}) - u_i(x)$ is continuous and $\mathcal{X}$ is compact, so there is a definite function $\eta \equiv \eta(\delta)$ such that if $\|x - x'\| \leq \eta(\delta)$, then $|f(x) - f(x')| \leq \delta$. Now, by assumption, $\{\forall x^* \in \mathcal{L}(X), u_i(z_i; x^*_{-i}) > u_i(x^*)\}$ is of positive probability. We thus get

$$0 < \mathbb{P}\big\{\forall x^* \in \mathcal{L}(X), u_i(z_i; x^*_{-i}) > u_i(x^*)\big\}$$

$$= \mathbb{P}\left\{\inf_{x^* \in \mathcal{L}(X)} \big(u_i(z_i; x^*_{-i}) - u_i(x^*)\big) > 0\right\} \tag{C.2a}$$

$$= \mathbb{P}\left(\bigcup_{m>0} \left\{\inf_{x^* \in \mathcal{L}(X)} \big(u_i(z_i; x^*_{-i}) - u_i(x^*)\big) > 2^{-m}\right\}\right) \tag{C.2b}$$

$$\leq \frac{1}{2}\mathbb{P}\big\{\forall x^* \in \mathcal{L}(X), u_i(z_i; x^*_{-i}) - u_i(x^*) > 2c\big\} \tag{C.2c}$$

for some $c > 0$ in (C.2c), and where (C.2a) is because $\mathcal{L}(X)$ is closed – hence compact – almost surely. Therefore, by definition of $\eta(\cdot)$,

$$0 < \mathbb{P}\{\forall x^* \in \mathcal{X}, \text{dist}(x^*, \mathcal{L}(X)) \leq \eta(c) \Rightarrow u_i(z_i; x_{-i}^*) - u_i(x^*) > c\} = 2\epsilon \qquad \text{(C.2d)}$$

Now, let $t_0$ such that $\mathbb{P}\{\forall t \geq t_0, \text{dist}(X_t, \mathcal{L}(X)) \leq \eta(c)\} > 1 - \frac{\epsilon}{2}$. Then by construction, we get

$$\mathbb{P}\{\forall t \geq t_0, u_i(z_i; X_{t,-i}) > u_i(X_t) + c\} > \epsilon. \qquad \text{(C.3)}$$

∎

and our proof is complete.

Intuitively, the existence of an action that consistently outperforms $X_t$ runs contrary to the behavior that one would expect from any regularized learning algorithm. We will proceed to make this intuition precise below by means of an energy argument. To that end, consider the Fenchel coupling

$$F_t = h_i(z_i) + h_i^*(Y_{i,t}) - \langle Y_{i,t}, z_i \rangle \qquad \text{(C.4)}$$

Then, by Lemma A.2 in Appendix A, we readily get that

$$F_{t+1} \leq F_t - \gamma_t \langle \hat{v}_{i,t}, z_i - X_{i,t} \rangle + \frac{\gamma_t^2}{2\kappa_h} \|\hat{v}_{i,t}\|_\infty^2. \qquad \text{(C.5)}$$

where, in obvious notation, we are identifying $z_i \in \mathcal{A}_i$ with the corresponding vertex $e_{z_i}$ of $\mathcal{X}_i = \Delta(\mathcal{A}_i)$. To proceed, the main idea will be to relate $\gamma_t \langle \hat{v}_{i,t}, z_i - X_{i,t} \rangle$ to its "perfect" counterpart $\gamma_t \langle v_i(X_t), z_i - X_{i,t} \rangle$. We formalize this below.

**Step 2.** *If $\mathcal{L}(X)$ is not resilient to $(i, z_i)$, there exists $t_1 \geq t_0$ such that, with probability $\varepsilon'/2 > 0$, and for all $t \geq t_1$, we have*

$$F_t \leq F_{t_0} - \frac{c}{2} \sum_{s=t_0}^{t} \gamma_s. \qquad \text{(C.6)}$$

*Proof of Step 2.* With probability $\varepsilon'$ and for all $t \geq t_0$, we have

$$\gamma_t \langle \hat{v}_{i,t}, z_i - X_{i,t} \rangle = \gamma_t \langle v_i(X_t), z_i - X_{i,t} \rangle + \gamma_t \langle U_{i,t}, z_i - X_{i,t} \rangle + \gamma_t \langle b_{i,t}, z_i - X_{i,t} \rangle \qquad \text{(C.7)}$$

$$\geq [c + \langle U_{i,t}, z_i - X_{i,t} \rangle + \langle b_{i,t}, z_i - X_{i,t} \rangle] \gamma_t. \qquad \text{(C.8)}$$

The combination of Eqs. (C.5) and (C.8) then provides the following upper bound of $F_{t+1}$:

$$F_{t+1} \leq F_t - c\gamma_t + \gamma_t \langle U_{i,t}, z_i - X_{i,t} \rangle + \gamma_t \langle b_{i,t}, z_i - X_{i,t} \rangle + \frac{\gamma_t^2}{2\kappa_h} \|\hat{v}_{i,t}\|_\infty^2 \qquad \text{(C.9)}$$

$$\leq F_{t_0} - c \sum_{s=t_0}^{t} \gamma_s + \underbrace{\sum_{s=t_0}^{t} \gamma_s \langle U_{s,i}, z_i - X_{s,i} \rangle}_{E_{U,t}} + \underbrace{\sum_{s=t_0}^{t} \gamma_s \langle b_{s,i}, z_i - X_{s,i} \rangle}_{E_{b,t}} + \sum_{s=t_0}^{t} \frac{\|\hat{v}_{s,i}\|_\infty^2}{2\kappa_h} \gamma_s^2. \qquad \text{(C.10)}$$

We are thus left to show is that $c \sum_{s=t_0}^{t} \gamma_s$ is the dominant term above. To do so, we proceed to examine each term individually:

- *Second-order term:* We first deal with the second-order term $\sum_{s=t_0}^{t} \frac{\|\hat{v}_{s,i}\|_\infty^2}{2\kappa_h} \gamma_s^2$. By expanding the $\|\hat{v}_{s,i}\|_\infty^2$, we readily get

$$\frac{\sum_{s=t_0}^{t} \|\hat{v}_{s,i}\|_\infty^2 \gamma_s^2}{\tau_t} = \mathcal{O}\left(\frac{\sum_{s=1}^{t} \gamma_s^2(1 + B_s^2 + \sigma_s^2)}{\sum_{s=1}^{t} \gamma_s}\right). \qquad \text{(C.11)}$$

However, by our assumptions on the parameters of (RL), we readily get

$$\lim_{t \to \infty} \frac{\gamma_t^2(1 + B_t^2 + \sigma_t^2)}{\gamma_t} = 0 \qquad \text{(C.12)}$$

so we conclude that

$$\lim_{t \to \infty} \frac{\sum_{s=1}^{t} \gamma_s^2(1 + B_s^2 + \sigma_s^2)}{\sum_{s=1}^{t} \gamma_s} = 0 \qquad \text{(C.13)}$$

by the Stolz-Cesàro theorem.

- *Bias term:* By far the most immediate, the bias term $E_{b,t}$ is bounded as

$$E_{b,t} \le 2 \sum_{s=t_0}^{t} \|b_{i,t}\|_\infty \gamma_s \le 2 \sum_{s=t_0}^{t} B_s \gamma_s = o\left( \sum_{s=t_0}^{t} \gamma_s \right) \quad \text{as } t \to \infty. \tag{C.14}$$

- *Noise term:* Finally, the noise term $E_{U,t}$ is bounded by means of the Azuma-Hoeffding inequality, cf. Lemma A.3 in Appendix A. Specifically, with probability at least $1 - \varepsilon'/2$, we have

$$\begin{aligned}
E_{U,t} &:= \sum_{s=t_0}^{t} \gamma_s \langle U_{s,i}, z_i - X_{s,i} \rangle \\
&\le 2 \left( \sum_{s=t_0}^{t} \|U_{s,i}\|_\infty^2 \gamma_s^2 \right)^{1/2} \sqrt{2 \log\left( \tfrac{4t^2}{\varepsilon'} \right)} \\
&\le 2 \left( \sum_{s=t_0}^{t} \sigma_s^2 \gamma_s^2 \right)^{1/2} \sqrt{2 \log\left( \tfrac{4t^2}{\varepsilon'} \right)}. \tag{C.15}
\end{aligned}$$

for all $t \ge t_0$. To proceed, note that a second application of the Stolz-Cesàro theorem yields $\sum_{s=t_0}^{t} \sigma_s^2 \gamma_s^2 = o(\sum_{s=t_0}^{t} \gamma_s)$ and, moreover, note that $\log(4t^2/\varepsilon') = \mathcal{O}(\sum_{s=t_0}^{t} \gamma_s)$. Taking square roots and multiplying then yields that

$$E_{U,t} = o\left( \sum_{s=t_0}^{t} \gamma_s \right) \tag{C.16}$$

with probability at least $1 - \varepsilon'/2$.

We are now in a position to establish the bound Eq. (C.6). Indeed, putting Eqs. (C.13), (C.14) and (C.16) together, we readily infer that there exists $t_1 \ge t_0$ such that, with probability at least $1 - \varepsilon'/2$, we have

$$\sum_{s=t_0}^{t} \gamma_s \langle U_{s,i}, z_i - X_{s,i} \rangle + \sum_{s=t_0}^{t} \gamma_s \langle b_{s,i}, z_i - X_{s,i} \rangle + \sum_{s=t_0}^{t} \frac{\|\hat{v}_{s,i}\|_\infty^2}{2\kappa_h} \gamma_s^2 \le \frac{c}{2} \sum_{s=t_0}^{t} \gamma_s \tag{C.17}$$

for all $t \ge t_1$. This proves Eq. (C.6) and concludes our proof. ∎

Summarizing the above, we have shown that, with probability at least $1 - \varepsilon'/2$, we have

$$F_{t+1} \le F_{t_0} - \frac{c}{2} \sum_{s=t_0}^{t} \gamma_s \to -\infty \quad \text{as } t \to \infty. \tag{C.18}$$

Since $F$ is nonnegative (by Lemma A.2), we have established that the event where $\mathcal{L}(X)$ is not resilient to $(i, z_i)$ is an event of probability zero. However, since there are uncountably many strategic deviations, the proof is not yet complete; the last step involves an approximation by deviations with *rational* entries.

**Step 3.** $\mathcal{L}(X)$ *is almost-surely resilient.*

*Proof of Step 3.* The key point of the proof is the observation that a closed set is resilient if and only if it is *rationally* resilient, i.e., it nullifies all *rational* deviations $z_i \in \mathcal{X}_i \cap \mathbb{Q}^{\mathcal{A}_i}$ (which are countably many). Indeed, if $\mathcal{L}(X)$ is not resilient with positive probability, then, likewise, $\mathcal{L}(X)$ will not be rationally resilient with positive probability either. Because there are countably many rational deviations, there must be a rational strategic deviation $(i, z_i)$ (with $z_i \in \mathcal{X}_i \cap \mathbb{Q}^{\mathcal{A}_i}$) to which $\mathcal{L}(X)$ is not resilient. This comes in contradiction with the conclusions of Step 2. ∎

This concludes the last required step, so the proof of Theorem 1 is now complete. ∎

# D    Proof of Theorems 2 and 3

In this last appendix, our goal is to prove our characterization of club sets, namely:

**Theorem 2.** *Fix some set $S \in \mathcal{P}(\mathcal{X})$ and suppose that (RL) is run with a steep regularizer and step-size / gain parameters $\ell_\gamma \in [0, 1]$, $\ell_b > 0$, and $\ell_\sigma < 1/2$. Then:*

1. *$S$ is stochastically asymptotically stable under (RL) if and only if it is a club set.*
2. *$S$ is irreducibly stable under (RL) if and only if it is an m-club set.*

**Theorem 3.** *Let $S \in \mathcal{P}(\mathcal{X})$ be a club set, and let $X_t$, $t = 1, 2, \ldots$, be the sequence of play generated by (RL) with parameters $\ell_\gamma \in [0, 1]$, $\ell_b > 0$, and $\ell_\sigma < 1/2$. Then, for all $\epsilon > 0$, there exists an (open, unbounded) initialization domain $\mathcal{D} \subseteq \mathcal{Y}$ such that, with probability at least $1 - \epsilon$, we have*

$$\mathrm{dist}(X_t, S) \leq C\varphi\Big(c_1 - c_2 \sum_{s=1}^t \gamma_s\Big) \quad \text{whenever } Y_1 \in \mathcal{D} \tag{17}$$

*where $C, c_1, c_2$ are constants ($C, c_2 > 0$), and the rate function $\varphi$ is given by $\varphi(z) = (\theta')^{-1}(z)$ if $z > \lim_{z \to 0^+} \theta'(z)$, and $\varphi(z) = 0$ otherwise.*

Our proof strategy will be to construct a sheaf of "linearized" energy functions which, when bundled together, yield a suitable Lyapunov-like function for $S$. To do so, let $\mathcal{C} = \prod_i \mathcal{C}_i$ denote the support of $S$ (cf. the definition of club sets), and let

$$\mathcal{Z}_i = \{e_{i\alpha_i'} - e_{i\alpha_i} : \alpha_i \in \mathcal{C}_i, \alpha_i' \in \mathcal{A}_i \setminus \mathcal{C}_i\} \tag{D.1}$$

and

$$\mathcal{Z} = \bigcup_{i \in \mathcal{N}} \mathcal{Z}_i \tag{D.2}$$

denote the set of all pure strategic deviations from $S$. Then, our ensemble of candidate energy functions will be given by

$$E_z(y) = \langle y, z \rangle \qquad \text{for } z \in \mathcal{Z}, y \in \mathcal{V}^*. \tag{D.3}$$

The motivation for this definition is given by the following lemma.

**Lemma D.1.** *Suppose that the sequence $y_t \in \mathcal{V}^*$, $t = 1, 2, \ldots$, has $E_z(y_t) \to -\infty$ for all $z \in \mathcal{Z}$ as $t \to \infty$. Then the sequence $x_t = Q(y_t)$ converges to $S$ as $t \to \infty$.*

*Proof.* Let $z = e_{i\alpha_i'} - e_{i\alpha_i}$ for some $i \in \mathcal{N}$, $\alpha_i \in \mathcal{C}_i$, and $\alpha_i' \in \mathcal{A}_i \setminus \mathcal{C}_i$. Since $E_z(y_t) \to -\infty$ by assumption, we get $y_{i\alpha_i',t} - y_{i\alpha_i,t} \to -\infty$ and hence, by Lemma A.1, we conclude that $Q_{i\alpha_i'}(x_t) \to 0$ as $t \to \infty$. In turn, given that this holds for all $i \in \mathcal{N}$ and all $\alpha_i' \in \mathcal{A}_i \setminus \mathcal{C}_i$, we conclude that $x_t = Q(y_t)$ converges to $S$. ∎

In view of the above, we will focus on showing that $E_z(Y_t) \to -\infty$ for all $z \in \mathcal{Z}$. As a first step, we establish a basic template inequality for the evolution of $E_z$ under (RL).

**Lemma D.2.** *Fix some $z \in \mathcal{Z}$ and let $E_t := E_z(Y_t)$. Then, for all $t = 1, 2, \ldots$, we have*

$$E_{t+1} \leq E_t + \gamma_t \langle v(X_t), z \rangle + \gamma_t \xi_t + \gamma_t \psi_t \tag{D.4}$$

*where the error terms $\xi_t$ and $\psi_t$ are given by*

$$\xi_t = \langle U_t, z \rangle \qquad \text{and} \qquad \psi_t = 2B_t. \tag{D.5}$$

*Proof.* Simply set $y \leftarrow Y_{t+1}$ in $E_z(y)$, invoke the definition of the update $Y_t \leftarrow Y_{t+1}$ in (RL), and note that $|\langle b_t, z \rangle| \leq \|z\| \|b_t\|_\infty \leq 2B_t$ by the definition of $\mathcal{Z}$. ∎

The key take-away from (D.4) is that, if $X_t$ is close to $S$ and $\alpha_i \in \mathcal{C}_i$, $\alpha_i' \in \mathcal{A}_i \setminus \mathcal{C}_i$, we will have

$$\langle v(X_t), z \rangle = v_{i\alpha_i'}(X_t) - v_{i\alpha_i}(X_t) = u_i(\alpha_i'; X_{-i,t}) - u_i(\alpha_i; X_{-i,t}) < 0 \tag{D.6}$$

by the continuity of $u_i$ and the assumption that $S$ is a club set. More concretely, by the definition of the better-reply correspondence, we have

$$\langle v(x^*), z \rangle < 0 \quad \text{for all } x^* \in S \text{ and all } z \in \mathcal{Z} \tag{D.7}$$

and hence, by continuity, there exists a neighborhood $\mathcal{B}$ of $\mathcal{S}$ such that

$$\langle v(x), z \rangle < 0 \quad \text{for all } x \in \mathcal{B} \text{ and all } z \in \mathcal{Z}. \tag{D.8}$$

In other words, as long as $X_t$ is sufficiently close to $\mathcal{S}$, (D.4) exhibits a consistent negative drift pushing $E_t$ towards $-\infty$.

To exploit this "dynamic consistency" property of $\mathcal{S}$, it will be convenient to introduce the family of sets

$$\mathcal{D}(\epsilon) = \{ y \in \mathcal{V}^* : \langle y, z \rangle < -\epsilon \text{ for all } z \in \mathcal{Z} \} \tag{D.9}$$

As we show below, these sets are mapped under $Q$ to neighborhoods of $\mathcal{S}$, so they are particularly well-suited to serve as initialization domains for (RL). This is encoded in the following properties:

**Lemma D.3.** *Let $x = Q(y)$ for some $y \in \mathcal{V}^*$. Then, for all $\alpha_i, \alpha_i', i \in \mathcal{N}$, we have*

$$x_{i\alpha_i} \leq \varphi\Big( \theta(1^-) + y_{i\alpha_i'} - y_{i\alpha_i} \Big) \tag{D.10}$$

*with $\varphi$ defined as per Theorem 3, i.e.,*

$$\varphi(z) = \begin{cases} 0 & \text{if } z \leq \theta'(0^+), \\ (\theta')^{-1}(z) & \text{if } \theta'(0^+) < z < \theta'(1^-), \\ 1 & \text{if } z \geq \theta'(1^-). \end{cases} \tag{D.11}$$

**Corollary D.1.** *For all $\delta > 0$ there exists some $\epsilon_\delta \in \mathbb{R}$ such that, for all $\epsilon > \epsilon_\delta$ and all $y \in \mathcal{D}_\epsilon$, we have*

$$Q_{i\alpha_i'}(y_i) < \delta \quad \text{for all } \alpha_i' \in \mathcal{A}_i \setminus \mathcal{C}_i \text{ and all } i \in \mathcal{N}. \tag{D.12}$$

*Proof of Lemma D.3.* Suppressing the player index for simplicity, the first-order stationarity conditions for the convex problem (9) readily give

$$y_\alpha - \theta'(x_\alpha) = \mu - \nu_\alpha, \tag{D.13}$$

where $\mu$ is the Lagrange multiplier for the equality constraint $\sum_\alpha x_\alpha = 1$, and $\nu_\alpha$ is the complementary slackness multiplier of the inquality constraint $x_\alpha \geq 0$ (so $\nu_\alpha = 0$ whenever $x_\alpha > 0$). Thus, rewriting (D.13) for some $\alpha \in \mathcal{A}$, we get

$$y_{\alpha'} - y_\alpha = \theta'(x_{\alpha'}) - \theta'(x_\alpha) + \nu_\alpha - \nu_{\alpha'} \tag{D.14}$$

and hence

$$\theta'(x_{\alpha'}) = \theta'(x_\alpha) + \nu_{\alpha'} - \nu_\alpha + y_{\alpha'} - y_\alpha \leq \theta'(1^-) + \nu_{\alpha'} + y_{\alpha'} - y_\alpha, \tag{D.15}$$

where we used the fact that $\nu_\alpha \geq 0$. Now, if $\theta'(1^-) + y_{\alpha'} - y_\alpha < \theta'(0^+)$ and $x_{\alpha'} > 0$ (so $\nu_{\alpha'} = 0$), we will have $\theta'(x_{\alpha'}) < \theta'(0^+)$, a contradiction. This shows that $x_{\alpha'} = 0$ if $\theta'(1^-) + y_{\alpha'} - y_\alpha < \theta'(0^+)$, so (D.10) is satisfied in this case. Otherwise, if $x_{\alpha'} > 0$, we must have $\nu_{\alpha'} = 0$ by complementary slackness, so (D.10) follows by applying the second branch of (D.11) to (D.15). ∎

The above provides us with a fairly good handle on the local geometric and dynamic properties of $\mathcal{S}$. On the flip side however, the various error terms in (D.5) may be positive, so $E_t$ may fail to be decreasing and $X_t$ may drift away from $\mathcal{S}$. On that account, it will be convenient to introduce the aggregate error processes

$$\mathrm{I}_t = \sum_{s=1}^t \gamma_s \xi_s \qquad \text{and} \qquad \mathrm{II}_t = \sum_{s=1}^t \gamma_s \psi_s. \tag{D.16}$$

Intuitively, the aggregates (D.16) measure the total effect of each error term in (D.4), so we will establish a first series of results under the following general requirements:

1. *Subleading error growth:*

$$\lim_{t \to \infty} \mathrm{I}_t / \tau_t = 0 \tag{Sub.I}$$

$$\lim_{t \to \infty} \mathrm{II}_t / \tau_t = 0 \tag{Sub.II}$$

where $\tau_t = \sum_{s=1}^t \gamma_s$ and both limits are to be interpreted in the almost sure sense.

2. *Drift dominance:*

$$\mathbb{P}(\mathrm{I}_t \leq C\tau_t^\alpha/2 \ \text{ for all } t) \geq 1 - \eta \tag{Dom.I}$$

$$\mathbb{P}(\mathrm{II}_t \leq C\tau_t^\alpha/2 \ \text{ for all } t) \geq 1 - \eta \tag{Dom.II}$$

for some $C > 0$ and $\alpha \in [0, 1)$.

In a nutshell, (Sub) posits that the aggregate error processes $\mathrm{I}_t$ and $\mathrm{II}_t$ of (D.16) are subleading relative to the long-run drift of (D.4), while (Dom) goes a step further and asks that said errors are asymptotically dominated by the drift in (D.4). Accordingly, under these implicit error control conditions, we obtain the interim convergence result below:

**Proposition D.1.** *Let $\mathcal{S}$ be a club set, fix some confidence threshold $\eta > 0$, and let $X_t = Q(Y_t)$ be the sequence of play generated by (RL). If (Sub) and (Dom) hold, there exists an unbounded initialization domain $\mathcal{D} \subseteq \mathcal{V}^*$ such that*

$$\mathbb{P}(X_t \text{ converges to } \mathcal{S} \mid Y_1 \in \mathcal{D}) \geq 1 - 2\eta. \tag{D.19}$$

*Proof of Proposition D.1.* Fix some $z \in \mathcal{Z}$, let $E_t = E_z(Y_t)$, and pick $\alpha \in [0, 1)$ so that (Dom) holds for some $C > 0$. In addition, set $c = -\sup_{x \in \mathcal{B}} \langle v(x), z \rangle > 0$, let $t_0 = \inf\{t : c\tau_t > C\tau_t^\alpha\}$, and write $\Delta E = \max_t \{C\tau_t^\alpha - c\tau_t\}$. Then, if $Y_1$ is initialized in $\mathcal{D} \leftarrow \mathcal{D}(\epsilon + \Delta E)$ where $\epsilon$ is such that $\mathcal{D}(\epsilon) \subseteq \mathcal{B}$, we will have $Y_t \in \mathcal{D}(\epsilon)$ for all $t$. Indeed, this being trivially the case for $t = 1$, assume it to be the case for all $s = 1, 2, \ldots, t$. Then, by (D.4) and our inductive hypothesis, we get

$$E_{t+1} \leq E_1 - \sum_{s=1}^t \gamma_s \langle v(X_s), z \rangle + \mathrm{I}_t + \mathrm{II}_t \leq -\epsilon - \Delta E - c\tau_t + C\tau_t^\alpha/2 + C\tau_t^\alpha/2 \leq -\epsilon - \Delta E + \Delta E = -\epsilon \tag{D.20}$$

i.e., $E_{t+1} \in \mathcal{D}(\epsilon)$, as claimed.

Now, since $E_t \in \mathcal{D}(\epsilon)$ for all $t$, we conclude that

$$E_{t+1} \leq E_1 - c\tau_t + \mathrm{I}_t + \mathrm{II}_t \quad \text{for all } t = 1, 2, \ldots \tag{D.21}$$

Thus, if (Sub) holds, we readily get $E_t \to -\infty$ with probability 1 on the event that (Dom.I) and (Dom.II) both hold. This implies that $E_t \to -\infty$, and since $z \in \mathcal{Z}$ above is arbitrary, we conclude that $X_t \to \mathcal{S}$ with probability at least $1 - 2\eta$, as claimed. ∎

We are now in a position to prove Theorem 2.

*Proof of Theorem 2.* Our proof will hinge on showing that (Sub) and (Dom) hold under the stated step-size and sampling parameter schedules. Our claim will then follow by a direct application of Proposition D.1 and a reduction to a suitable subface of $\mathcal{X}$.

First, regarding (Sub), the law of large numbers for martingale difference sequences [25, Theorem 2.18] shows that $\mathrm{I}_t/\tau_t \to 0$ with probability 1 on the event $\left\{ \sum_t \gamma_t^2 \mathbb{E}[\xi_t^2 \mid \mathcal{F}_t]/\tau_t^2 < \infty \right\}$. However

$$\mathbb{E}[\xi_t^2 \mid \mathcal{F}_t] \leq 2^2 \, \mathbb{E}[\|U_t\|_\infty^2 \mid \mathcal{F}_t] \leq 2^2 \sigma_t^2 = \mathcal{O}(t^{2\ell_\sigma}) \tag{D.22}$$

so, in turn, we get

$$\sum_t \frac{\gamma_t^2 \, \mathbb{E}[\xi_t^2 \mid \mathcal{F}_t]}{\tau_t^2} = \mathcal{O}\!\left( \sum_t \frac{\gamma_t^2 \sigma_t^2}{\tau_t^2} \right) = \mathcal{O}\!\left( \sum_t \frac{t^{-2\ell_\gamma} t^{2\ell_\sigma}}{t^{2(1-\ell_\gamma)}} \right) = \mathcal{O}\!\left( \sum_t \frac{1}{t^{2-2\ell_\sigma}} \right) < \infty \tag{D.23}$$

given that $\ell_\sigma < 1/2$. This establishes (Sub.I); the remaining requirement (Sub.II) follows trivially by noting that $\sum_{s=1}^t \gamma_s B_s \big/ \sum_{s=1}^t \gamma_s \to 0$ if and only if $B_t \to 0$, which is immediate from the theorem's assumptions.

Second, regarding (Dom), since $B_t$ is deterministic and $B_t = \mathcal{O}(1/t^{\ell_b})$ for some $\ell_b > 0$, it is always possible to find $C > 0$ and $\alpha \in (0, 1)$ so that (Dom.II) holds. We are thus left to establish (Dom.I). To that end, let $\mathrm{I}_t^* = \sup_{1 \leq s \leq t} |\mathrm{I}_t|$ and set $P_t := \mathbb{P}\!\left( \mathrm{I}_t^* > C\tau_t^\alpha/2 \right)$ so

$$P_t \leq \frac{\mathbb{E}[|\mathrm{I}_t|^q]}{(C/2)^q \tau_t^{\alpha q}} \leq c_q \frac{\mathbb{E}\!\left[ \left( \sum_{s=1}^t \gamma_s^2 \|U_s\|_\infty^2 \right)^{q/2} \right]}{\tau_t^{\alpha q}} \tag{D.24}$$

where $c_q$ is a positive constant depending only on $C$ and $q$, and we used Kolmogorov's inequality (Lemma A.4) in the first step and the Burkholder–Davis–Gundy inequality (Lemma A.6) in the second.

To proceed, we will require the following variant of Hölder's inequality [9, p. 15]:

$$\left(\sum_{s=1}^{t} a_s b_s\right)^{\rho} \leq \left(\sum_{s=1}^{t} a_s^{\frac{\lambda\rho}{\rho-1}}\right)^{\rho-1} \sum_{s=1}^{t} a_s^{(1-\lambda)\rho} b_s^{\rho} \tag{D.25}$$

valid for all $a_s, b_s \geq 0$ and all $\rho > 1$, $\lambda \in [0, 1)$. Then, substituting $a_s \leftarrow \gamma_s^2$, $b_s \leftarrow \|U_s\|_\infty^2$, $\rho \leftarrow q/2$ and $\lambda \leftarrow 1/2 - 1/q$, (D.24) gives

$$P_t \leq c_q \frac{\left(\sum_{s=1}^{t} \gamma_s\right)^{q/2-1} \sum_{s=1}^{t} \gamma_s^{1+q/2} \mathbb{E}[\|U_s\|_\infty^q]}{\tau_t^{\alpha q}} \leq c_q \frac{\sum_{s=1}^{t} \gamma_s^{1+q/2} \sigma_s^q}{\tau_t^{1+(\alpha-1/2)q}} \tag{D.26}$$

We now consider two cases, depending on whether the numerator of (D.26) is summable or not.

*Case 1:* $\ell_\gamma(1+q/2) \geq 1+q\ell_\sigma$. In this case, the numerator of (D.26) is summable under the theorem's assumptions, so the fraction in (D.26) behaves as $\mathcal{O}(1/t^{(1-\ell_\gamma)(1+(\alpha-1/2)q)})$.

*Case 2:* $\ell_\gamma(1 + q/2) < 1 + q\ell_\sigma$. In this case, the numerator of (D.26) is not summable under the theorem's assumptions, so the fraction in (D.26) behaves as $\mathcal{O}\left(t^{1-\ell_\gamma(1+q/2)+q\ell_\sigma} \big/ t^{(1-\ell_\gamma)(1+(\alpha-1/2)q)}\right)$.

Thus, working out the various exponents, a tedious – but otherwise straightforward – calculation shows that there exists some $\alpha \in (0, 1)$ such that $P_t$ is summable as long as $\ell_\sigma < 1/2 - 1/q$ and $0 \leq \ell_\gamma < q/(2+q)$. Hence, if $\gamma$ is sufficiently small relative to $\eta$, we conclude that

$$\mathbb{P}(I_t \leq C\tau_t^\alpha/2 \text{ for all } t) \geq 1 - \textstyle\sum_t P_t \geq 1 - \eta/2. \tag{D.27}$$

Finally, if $\ell_\gamma > 1/2 + \ell_\sigma$, (Dom.I) is a straightforward consequence of (D.24) for $q = 2$.

With all this in hand, the final steps of our proof proceed as follows:

**Closedness $\implies$ Stability.**  Our assertion follows by invoking Proposition D.1. ∎

**Stability $\implies$ Closedness.**  Suppose that $\mathcal{S}$ is not club. Then there exists some pure strategy $\alpha \in \mathcal{C}$ and some deviation $\alpha' \notin \mathcal{C}$ such that the deviation from $\alpha$ to $\alpha'$ is not costly to the deviating player. Thus, if we consider the restriction of the game to the face spanned by $\alpha$ and $\alpha'$ (a single-player game with two strategies), the corresponding score difference will be

$$y_{\alpha',t} - y_{\alpha,t} \geq \sum_{s=1} \gamma_s b_s + \sum_{s=1} \gamma_s U_s \tag{D.28}$$

By our standing assumptions for $b_t$ and $U_t$ (and Doob's martingale convergence theorem for the latter), both $\sum_{s=1} \gamma_s b_s$ and $\sum_{s=1} \gamma_s U_s$ will be bounded from below by some (a.s.) finite random variable $A_0$. Since $\theta$ is steep, it follows that, with probability 1, $\liminf_{t\to\infty}(y_{\alpha,t}) > 0$, so $\mathcal{C}$ cannot be stable. ∎

**Minimality $\implies$ Irreducible Stability.**  Suppose that $\mathcal{S}$ is m-club. Then, by our previous claim, $\mathcal{S}$ is stochastically asymptotically stable. If $\mathcal{S}$ contains a proper subface $\mathcal{S}' \subsetneq \mathcal{S}$ that is also stochastically asymptotically stable, $\mathcal{S}'$ must be club by the converse implication of the first part of the theorem. However, in that case, $\mathcal{S}$ would not be m-club, a contradiction which proves our claim. ∎

**Irreducible Stability $\implies$ Minimality.**  For our last claim, assume that $\mathcal{S}$ is irreducibly stable. By the first part of our theorem, this implies that $\mathcal{S}$ is club. Then, if it so happens that $\mathcal{S}$ is not m-club, it would contain a proper club subface $\mathcal{S}' \subsetneq \mathcal{S}$; by the first part of our theorem, this set would be itself stochastically asymptotically stable, in contradiction to the irreducibility assumption. This shows that $\mathcal{S}$ is m-club and concludes our proof. ∎

We are only left to establish the convergence rate estimate of Theorem 3.

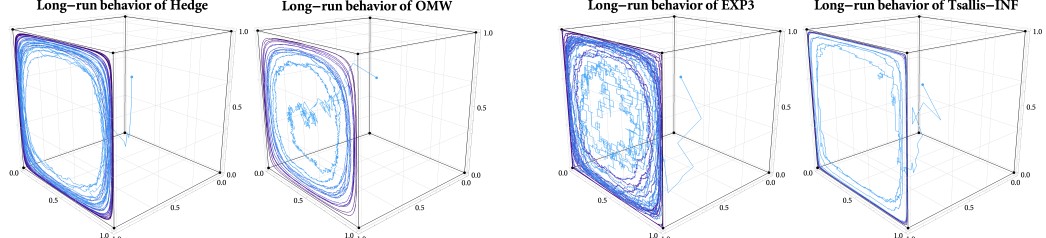

**Figure 2:** The long-run behavior of Algorithms 1–3 in a $2 \times 2 \times 2$ game. Algorithms 1 and 2 were run with a logit choice map as per (HEDGE); Algorithm 3 was run with both variants, EXP3 and TSALLIS-INF. All algorithms were run for $5 \times 10^5$ iterations with $\gamma_t = 1/t^{0.4}$ and $\delta_t = 0.1/t^{0.15}$; color indicates time, with darker hues indicating later iterations. The face to the left is closed under better replies, so $X_t$ converges quickly to said face (as per Theorems 2 and 3).

*Proof of Theorem 3.* Going back to (D.21) and invoking Lemma D.3 shows that there exist constants $c_1 > 0$ and $c_2 \in \mathbb{R}$ such that, for all $\alpha_i \in \mathcal{A}_i \setminus \mathcal{C}_i, i \in \mathcal{N}$, we have

$$X_{i\alpha_i, t} \leq \varphi(\theta(1^-) + E_t) \leq \varphi(c_2 - c_1 \tau_t) \tag{D.29}$$

with probability 1 on the events of (Dom). We thus get

$$\mathrm{dist}_1(X_t, \mathcal{S}) \leq \sum_{i \in \mathcal{N}} \sum_{\alpha_i \in \mathcal{A}_i \setminus \mathcal{C}_i} \varphi(c_2 - c_1 \tau_t), \tag{D.30}$$

and our proof is complete. ∎

As for the rate estimates of Corollary 2, the proof boils down to a simple derivation of the corresponding rate functions:

*Proof of Corollary 2.* By a straightforward calculation, we have:

1. If $\theta(z) = z \log z$ then $\varphi(z) = \exp(1 + z)$.
2. If $\theta(z) = -4\sqrt{z}$ then $\varphi(z) = 4/z^2$.
3. If $\theta(z) = z^2/2$ then $\varphi(z) = [z]_0^1$.

Our claims then follow immediatly from the rate estimate (17) of Theorem 2. ∎

# E   Numerical experiments

In all our experiments, we ran the EXP3 variant of bandit FTRL (B-FTRL) (cf. Algorithm 3) with step-size and sampling radius parameters $\gamma_t = 0.2 \times t^{-1/2}$ and $\delta_t = 0.1 \times t^{-0.15}$ respectively. The algorithm was run for $T = 10^4$ iterations and, to reduce graphical clutter, we plotted only every third point of each trajectory. Trajectories have been colored throughout with darker hues indicating later times (e.g., light blue indicates that the trajectory is closer in time to its starting point, darker shades of blue indicate proximity to the termination time). The algorithm's initial conditions were taken from a uniform initialization grid of the form $y_1 \in \{-1, 0, 1\}^3$ and perturbed by a uniform random number in $[-0.1, -0.1]$ to avoid non-generic initializations.

In general, the two defining elements of (RL) are *a*) the regularizer of the method; and *b*) the feedback available to the players. From our experiments, we conclude that methods with Euclidean regularization tend to have faster identification rates (i.e., converge to the support of an equilibrium / club set faster), but they are more "extreme" than methods with an "entropy-like" regularizer (in the sense that players tend to play pure strategies more often). As for the feedback available to the players, payoff-based methods tend to have higher variance (and hence a slower rate of convergence) relative to methods with full information; otherwise, from a qualitative viewpoint, there are no perceptible differences in their limiting behavior.

Finally, optimistic / extra-oracle methods with full information exhibit better convergence properties in two-player zero-sum games (relative to standard FTRL policies); however, this is a fragile advantage

that evaporates in the presence of noise and/or uncertainty (in which case "vanilla" and "optimistic" methods are essentially indistinguishable). We illustrate these findings in Fig. 2.

Regarding Fig. 1, the payoffs of the chosen games were normalized to $[-1, 1]$ and players are assumed to choose between two actions labeled "$O$" and "1". The specific tableaus are shown in the table below, next to the respective portrait (all taken from Fig. 1.

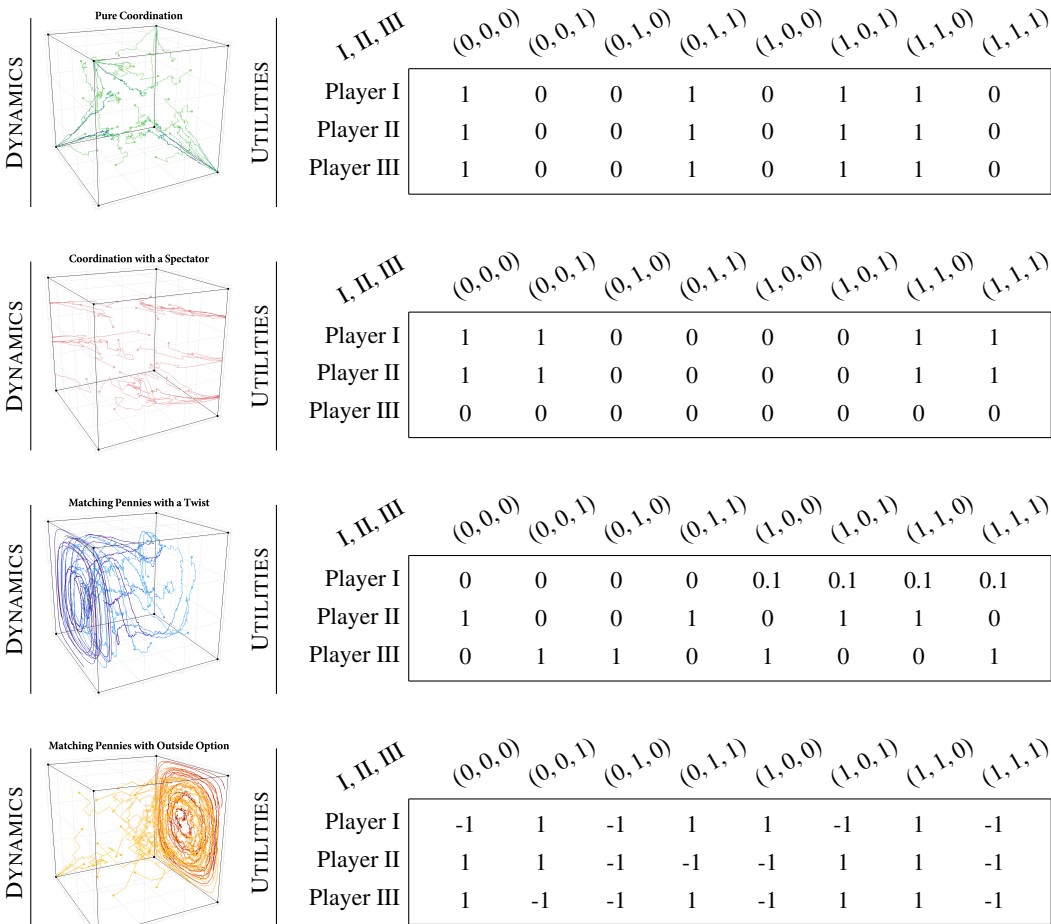

| I, II, III | $(0,0,0)$ | $(0,0,1)$ | $(0,1,0)$ | $(0,1,1)$ | $(1,0,0)$ | $(1,0,1)$ | $(1,1,0)$ | $(1,1,1)$ |
|---|---|---|---|---|---|---|---|---|
| **Pure Coordination** | | | | | | | | |
| Player I | 1 | 0 | 0 | 1 | 0 | 1 | 1 | 0 |
| Player II | 1 | 0 | 0 | 1 | 0 | 1 | 1 | 0 |
| Player III | 1 | 0 | 0 | 1 | 0 | 1 | 1 | 0 |
| **Coordination with a Spectator** | | | | | | | | |
| Player I | 1 | 1 | 0 | 0 | 0 | 0 | 1 | 1 |
| Player II | 1 | 1 | 0 | 0 | 0 | 0 | 1 | 1 |
| Player III | 0 | 0 | 0 | 0 | 0 | 0 | 0 | 0 |
| **Matching Pennies with a Twist** | | | | | | | | |
| Player I | 0 | 0 | 0 | 0 | 0.1 | 0.1 | 0.1 | 0.1 |
| Player II | 1 | 0 | 0 | 1 | 0 | 1 | 1 | 0 |
| Player III | 0 | 1 | 1 | 0 | 1 | 0 | 0 | 1 |
| **Matching Pennies with Outside Option** | | | | | | | | |
| Player I | -1 | 1 | -1 | 1 | 1 | -1 | 1 | -1 |
| Player II | 1 | 1 | -1 | -1 | -1 | 1 | 1 | -1 |
| Player III | 1 | -1 | -1 | 1 | -1 | 1 | 1 | -1 |

