# Strategic stability under regularized learning in games

## Abstract

In this paper, we examine the long-run behavior of regularized, no-regret learning in finite games. A well-known result in the field states that the empirical frequencies of no-regret play converge to the game's set of coarse correlated equilibria; however, our understanding of how the players' *actual* strategies evolve over time is much more limited – and, in many cases, non-existent. This issue is exacerbated by a series of recent results showing that *only* strict Nash equilibria are stable and attracting under regularized learning, thus making the relation between learning and pointwise solution concepts particularly elusive. In lieu of this, we take a more general approach and instead seek to characterize the *setwise* rationality properties of the players' day-to-day play. To that end, we focus on one of the most stringent criteria of setwise strategic stability, namely that any unilateral deviation from the set in question incurs a cost to the deviator – a property known as *closedness under better replies* (club). In so doing, we obtain a remarkable equivalence between strategic and dynamic stability: *a product of pure strategies is closed under better replies if and only if its span is stable and attracting under regularized learning.* In addition, we estimate the rate of convergence to such sets, and we show that methods based on entropic regularization (like the exponential weights algorithm) converge at a geometric rate, while projection-based methods converge within a *finite* number of iterations, even with bandit, payoff-based feedback.

## 1  Introduction

**Background.**    The question of whether players can learn to emulate rational behavior through repeated interactions has been one of the mainstays of non-cooperative game theory – and it has recently gained increased momentum owing to a surge of breakthrough applications to machine learning and data science (from online advertising to auctions and multi-agent reinforcement learning). Informally, this question can be stated as follows:

> *If each player follows an iterative procedure aiming to increase their individual payoff,*
> *does the players' long-run behavior converge to a rationally admissible state?*

A natural setting for studying this question is to assume that each player is following a no-regret algorithm, i.e., a policy which is asymptotically as good against a given sequence of payoff functions as the best fixed strategy in hindsight. In this framework, the link between learning and rationality is provided by a folk result which states that, under no-regret learning, the empirical frequency of play converges to the game's set of *coarse correlated equilibria* (CCE) – also known as the game's *Hannan set* [22]. This result has been of seminal importance to the field because no-regret play can be achieved via a wide class of "regularized learning" policies, as exemplified by the *"follow-the-regularized-leader"* (FTRL) family of algorithms [41, 42] and its variants – optimistic mirror descent [13, 36, 37, 43], HEDGE / EXP3 [4, 5, 9, 10], implicitly normalized forecasters [1, 3], etc.

All these policies have (at least) one thing in common: they seek to provide the tightest possible guarantees for each player's individual regret, thus accelerating convergence to the game's Hannan set. As such, in games where the marginalization of CCE coincides with the game's Nash equilibria (like two-player zero-sum games), we obtain a positive equilibrium convergence guarantee: the long-run average frequency of play evolves "as if" the players were rational to begin with – i.e., as if they had full knowledge of the game, common knowledge of rationality, the ability to communicate this knowledge, etc.

Submitted to 37th Conference on Neural Information Processing Systems (NeurIPS 2023). Do not distribute.

On the other hand, in many concrete applications – and, in particular, in the context of regularized learning –

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

**Pure Coordination**

| I, II, III | (0,0,0) | (0,0,1) | (0,1,0) | (0,1,1) | (1,0,0) | (1,0,1) | (1,1,0) | (1,1,1) |
|---|---|---|---|---|---|---|---|---|
| Player I | 1 | 0 | 0 | 1 | 0 | 1 | 1 | 0 |
| Player II | 1 | 0 | 0 | 1 | 0 | 1 | 1 | 0 |
| Player III | 1 | 0 | 0 | 1 | 0 | 1 | 1 | 0 |

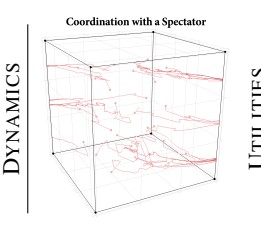

**Coordination with a Spectator**

| I, II, III | (0,0,0) | (0,0,1) | (0,1,0) | (0,1,1) | (1,0,0) | (1,0,1) | (1,1,0) | (1,1,1) |
|---|---|---|---|---|---|---|---|---|
| Player I | 1 | 1 | 0 | 0 | 0 | 0 | 1 | 1 |
| Player II | 1 | 1 | 0 | 0 | 0 | 0 | 1 | 1 |
| Player III | 0 | 0 | 0 | 0 | 0 | 0 | 0 | 0 |

DYNAMICS

**Matching Pennies with a Twist**

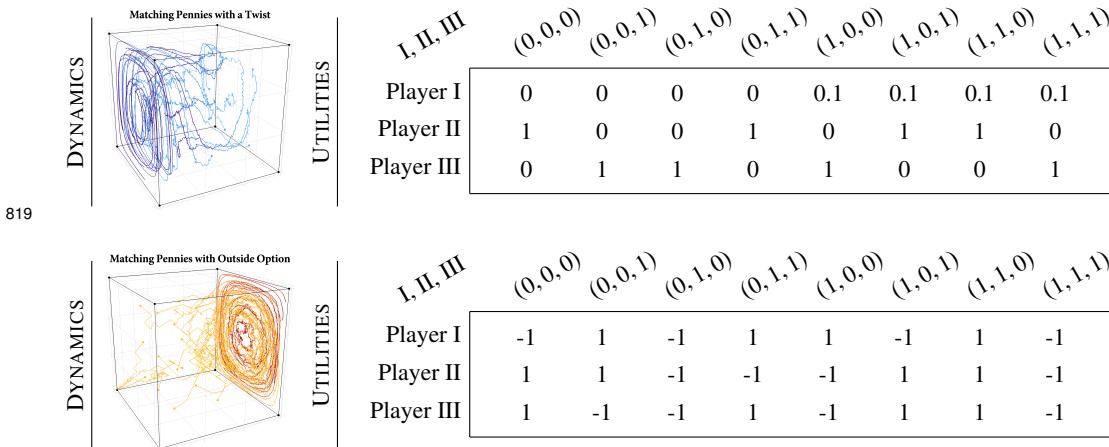

UTILITIES

| I, II, III | (0,0,0) | (0,0,1) | (0,1,0) | (0,1,1) | (1,0,0) | (1,0,1) | (1,1,0) | (1,1,1) |
|---|---|---|---|---|---|---|---|---|
| Player I | 0 | 0 | 0 | 0 | 0.1 | 0.1 | 0.1 | 0.1 |
| Player II | 1 | 0 | 0 | 1 | 0 | 1 | 1 | 0 |
| Player III | 0 | 1 | 1 | 0 | 1 | 0 | 0 | 1 |

819

DYNAMICS

**Matching Pennies with Outside Option**

UTILITIES

| I, II, III | (0,0,0) | (0,0,1) | (0,1,0) | (0,1,1) | (1,0,0) | (1,0,1) | (1,1,0) | (1,1,1) |
|---|---|---|---|---|---|---|---|---|
| Player I | -1 | 1 | -1 | 1 | 1 | -1 | 1 | -1 |
| Player II | 1 | 1 | -1 | -1 | -1 | 1 | 1 | -1 |
| Player III | 1 | -1 | -1 | 1 | -1 | 1 | 1 | -1 |