# OpenReview forum: "The Equivalence of Dynamic and Strategic Stability under Regularized Learning in Games"
_NeurIPS.cc/2023/Conference — NeurIPS 2023 spotlight_

### Official Review · Reviewer_vfB1 · 2023-07-03

**Soundness:** 3 good
**Presentation:** 3 good
**Contribution:** 3 good
**Rating:** 7
**Confidence:** 3

**Summary:**

This paper studies the long run behavior of no-regret learning, and introduces the notion of resilience to strategic deviations as a metric with which to characterize no-regret learning algorithms. Moreover, to further strengthen their results the paper utilizes the idea of setwise strategic stability (m-club). In particular, the paper shows a nice connection between regularized learning and club sets. Finally, they estimate convergence rates to club sets, showing geometric rate for entropic regularizers and finite iterations for projection-based methods.

**Strengths:**

I think this paper is a conceptual step forward in the area of regularized learning dynamics. A major point of contention in many prior works in learning in games has been the disconnect between desirable equilibria and equilibria that are actually converged to by regularized learning. The results given in the paper are fairly broad, and the characterization of club sets as an alternative solution concept for regularized learning is (as far as I can tell) quite watertight and intuitive.

**Weaknesses:**

I only have very minor complaints with the paper, namely the section on regularized learning that introduces a few different algorithms under the RL umbrella. I feel this section is quite long and the notation is unnecessarily heavy, most of these could be in the appendix. I would have preferred to see more space allocated to either an extended proof sketch for Thms 2 & 3, or more experimental details.

**Questions:**

A question regarding the actual convergence properties of regularized learners to club sets in comparison to classical solution concepts:  how do the payoffs of these points compare to the Nash equilibrium values? Do you see any interesting behaviors of note between standard FTRL dynamics and other RL dynamics that are significantly different in your experiments?

Also, is there any connection between club sets and the chain recurrent set of the dynamics? Of course the paper is focused on discrete time regularized learning, but do you have any intuition as to whether club sets could be useful to construct/converge to chain recurrent sets in the continuous-time setting?

**Limitations:**

The limitations and scope of the proposed ideas are discussed adequately.

---

> ### Author Rebuttal · Authors · 2023-08-09
>
> We are sincerely grateful for your detailed input and positive evaluation. We reply to your questions below and we will revise our manuscript accordingly at the next revision opportunity:
>
> 1. "*I feel Section 3 is quite long and the notation is unnecessarily heavy, most of these could be in the appendix. I would have preferred to see more space allocated to either an extended proof sketch for Thms 2 & 3, or more experimental details.*"
>
>    **Reply.** We understand your concern but, at the same time, we were constrained by the NeurIPS page limitations, and we had to do some tough choices in terms of presentation along the way. In this regard, we intend to take full advantage of the extra page allowed in the revision phase to implement the following restructuring changes:
>    - Include a quick "warm-up" section before the current Section 3, intended to discuss some basic algorithms and feedback models that are standard in the field (exponential weights, optimistic exponential weights, and bandit exponential weights).
>    - Proceed to state the "umbrella" FTRL template and quickly explain how it includes these algorithms as special cases (without any details).
>    - Relegate the remaining technical elements, definitions and examples to the appendix (in order to streamline the flow of the discussion), as per your suggestion.
>
>    This will provide the necessary examples and anchor points to motivate the general analysis and ease notation, and it will also leave us sufficient space to expand on the proof sketches of Theorems 1-3 (by bringing forward the step-by-step skeleton from the paper's current appendices).
>
> ---
>
> 2. "*How do the payoffs of these points compare to the Nash equilibrium values?*"
>
>    **Reply.** Each club set contains an essential component of Nash equilibria so, by the multilinearity of the players' payoff functions, the payoff values of the latter will be contained in the convex hull of the former (and, in the case of m-club sets, it cannot be contained in a smaller subset thereof). Without further structural assumptions on the underlying game, we are not aware of a finer relation between them.
>
> ---
>
> 3. "*Do you see any interesting behaviors of note between standard FTRL dynamics and other RL dynamics that are significantly different in your experiments?*"
>
>    **Reply.** In general, the two driving factors are (*a*) the regularizer of the method; and (*b*) the feedback available to the players. Methods with Euclidean regularization tend to have faster identification rates (i.e., converge to the support of an equilibrium / club set faster), but they are more "extreme" than methods with an "entropy-like" regularizer (in the sense that players tend to play pure strategies more often). As for the feedback available to the players, payoff-based methods tend to have higher variance (and hence a slower rate of convergence) relative to methods with full information; otherwise however, from a qualitative viewpoint, there are no perceptible differences in their limiting behavior. Finally, optimistic / extra-oracle methods with **full** information exhibit better convergence properties in two-player zero-sum games (relative to standard FTRL policies); however, this is a fragile advantage that evaporates in the presence of noise and/or uncertainty (in which case "vanilla" and "optimistic" methods are essentially indistinguishable).
>
>    Figure 2 in Appendix B was intended to illustrate a part of these findings, and we would be happy to expand on it and bring it to the main text if you think it would make things clearer for the reader.
>
> ---
>
> 4. "*Is there any connection between club sets and the chain recurrent set of the dynamics? Of course the paper is focused on discrete time regularized learning, but do you have any intuition as to whether club sets could be useful to construct/converge to chain recurrent sets in the continuous-time setting?*"
>
>    **Reply.** Excellent question! Due to space limitations, we could not expand on the continuous-time implications of our work but, indeed, there is an important relation between sets that are *minimally* closed under better replies (m-club) and sets that are internally chain recurrent.
>
>    First, in continuous time, the dynamics of regularized learning are described by the system $\dot y(t) = v(x(t))$ with $x(t) = Q(y(t))$, as per reference [28] of our paper. In this context, the analogue of Theorem 2 would state that a set is m-club if and only if it is irreducibly stable under the above dynamics: we do not prove this result, but our techniques can be used to show that this statement holds.
>
>    Given this equivalence, it follows that m-club sets are compact, invariant, and do not admit any asymptotically stable subsets with smaller support. This is not exactly the same as not admitting any proper *attractors*, but it is close - and since a set is internally chain recurrent if and only if it is compact, invariant, and does not admit any proper attractors, this would suggest that m-club sets are, in many cases, internally chain recurrent.
>
>    We conjecture that this relation is actually true in general, but we do not have a proof for this fact, and we believe it is a very fruitful direction for future research.
>
> ---
>
> We hope and trust that the above points address your questions - thank you again for your detailed input and positive evaluation!
>
> Kind regards,
>
> The authors

---

> > ### Comment · Reviewer_vfB1 · 2023-08-15
> > **Response to Author Rebuttal**
> >
> > Thank you very much for your detailed response. The suggestions for content in the additional page are very reasonable, and should make the paper more suitable for the NeurIPS audience.
> >
> > The comment about Figure 2 and comparisons to other RL methods is interesting but might seem a bit out of place without changes to the flow of the paper, though I think having more discussion about the figure in the appendix would suffice.
> >
> > Regarding chain recurrence, I agree that it is a fascinating connection! It is a shame that this cannot be expanded upon given space constraints but I am eagerly awaiting a future work in this direction.
> >
> > Best regards,
> > Reviewer vfB1

---

### Official Review · Reviewer_7Hjy · 2023-07-03

**Soundness:** 3 good
**Presentation:** 3 good
**Contribution:** 3 good
**Rating:** 6
**Confidence:** 3

**Summary:**

The paper shows the convergence of regularized learning in games to sets satisfying a property called closeness under better replies. Moreover, convergence rates are derived even with bandit feedback.

**Strengths:**

The pointwise behavior of regularized learning in games has gained lots of attention recently, and the paper considers a fundamental question in this field. The authors provide complete and general answers to the questions.

**Weaknesses:**

The paper is super notation-heavy and considers a very general setting in terms of both algorithms and feedback models. It would be much easier to follow if the authors could first provide some basic or toy examples and then generalize. Minor: a corrupted reference at the beginning of Line 267.

**Questions:**

While the questions (line 59-60) are important and the results are mathematically sound, what do they imply to regularized learning, in particular, for practitioners? For example, from prior works, we already know that a day-to-day strategy can be arbitrarily bad so it is probably better to do some averaging. Does this paper extend our understanding in this direction?

---

> ### Author Rebuttal · Authors · 2023-08-09
>
> Thank you for your detailed input and positive evaluation. We reply to your questions below and we will revise our manuscript accordingly at the next revision opportunity:
>
> 1. "*The paper is [notation-heavy and considers a very general setting] It would be much easier to follow if the authors could first provide some basic or toy examples and then generalize.*"
>
>    **Reply.** We understand your concern but, at the same time, we were constrained by the NeurIPS page limitations, and we had to do some tough choices in terms of presentation along the way. In this regard, we intend to take full advantage of the extra page allowed in the revision phase to implement the following restructuring changes:
>    - Include a quick "warm-up" section before the current Section 3, intended to discuss some basic algorithms and feedback models that are standard in the field (exponential weights, optimistic exponential weights, and bandit exponential weights).
>    - Proceed to state the "umbrella" FTRL template and quickly explain how it includes these algorithms as special cases (without any technical details that could distract the reader).
>    - Relegate the remaining technical elements and definitions to the appendix (in order to streamline the flow of the discussion), as per your suggestion.
>
>    This will provide the necessary examples and anchor points to motivate the general analysis and ease notation, so we trust and hope it addresses your notation concerns.
>
> ---
>
> 2. "*Minor: a corrupted reference at the beginning of Line 267.*"
>
>    **Reply.** Apologies, this was a broken reference to Appendix C - thanks for catching it.
>
> ---
>
> 3. "*From prior works, we already know that a day-to-day strategy can be arbitrarily bad so it is probably better to do some averaging. Does this paper extend our understanding in this direction?*"
>
>    **Reply.** There are several remarks to be made here, so we proceed point-by-point:
>    - First, from a practical point of view, when agents are involved in a real-time, online learning process (e.g., commuting from home to work each day), the payoff they obtain at each epoch is determined by the action they actually *played* at said epoch. In this context, an "averaged" sequence of strategies (either time-averages or empirical frequencies) is not as meaningful, because it is never actually *played* by the agents. As such, the day-to-day sequence of play becomes the de facto figure of merit for the problem, as it describes what the agents actually play and determines their in-game rewards.
>    - Second, if the game is not monotone (in the sense of operator theory and variational inequalities), the time-averaged sequence $\bar x_{i,t} = (1/t) \sum_{s=1}^t x_{i,s}$ has no convergence guarantees in general. The only general class of finite games which *is* monotone is two-player, zero-sum games: in this case, time-averaging can be beneficial as a technique for the **offline** computation of Nash equilibria but, even then, since $\bar x_{i,t}$ is never actually played by the players, it is not as meaningful from an **online** viewpoint.
>    - Finally, concerning the empirical frequency of play $\bar z_{\alpha,t} = (1/t)\sum_{s=1}^t \mathbb{I}(\alpha_t = \alpha)$ (which is not the same as the time-averaged sequence $\bar x_t$ above), it is indeed well known that, under no-regret learning, $\bar z_{t}$ converges to the Hannan set of the game (its set of CCE). However, as we discuss in Section 4, the Hannan set could contain highly non-rationalizable outcomes, e.g., with players selecting dominated strategies for all time (the counterexample of Viossat and Zapechelnyuk). Our result serves to exclude such outcomes by showing that the *only* part of the Hannan set which is stable and attracting under regularized learning is its intersection with the set of correlated actions that are supported on a **club** set - i.e., which is closed under better replies, and hence strategically stable.
>
>    We did not include a detailed discussion along those lines because, as we explained above, averaging in an online context is less relevant than in the offline case. However, we will be happy to take advantage of the extra page provided in the camera-ready phase to include the above discussion.
>
> ---
>
> We hope and trust that these points address your questions - thank you again for your input and positive evaluation!
>
> Kind regards,
>
> The authors

---

> > ### Comment · Reviewer_7Hjy · 2023-08-18
> >
> > I thank the authors for their response and confirm my positive evaluation.

---

### Official Review · Reviewer_nBFX · 2023-07-05

**Soundness:** 3 good
**Presentation:** 3 good
**Contribution:** 3 good
**Rating:** 7
**Confidence:** 3

**Summary:**

The paper deals with a fundamental question about long-term behavior of regularized learning algorithms in finite player finite action static games. They prove an interesting equivalence between the set of strategies which are closed under better replies with that of stochastically stable and attracting fixed points of wide class of regularized learning algorithm popularly studied in literature. Furthermore, they also study the rate of convergence of regularized learning algorithms to this set.



**Strengths:**

The paper studies a very fundamental equivalence relation between payoff structure of a static game with that of limit sets of regularized learning algorithms. This result is very strong result which enhances our understanding of learning in static games. Particuarly, this paper brings the notion of better replies from economic literature and presents a deep connection with asymptotic properties of learning algorithm.

The clarity of presentation of this paper is very good.

The literature survey is also up to the mark to the best of my knowledge.



**Weaknesses:**

The paper has no major weakness in my view.



**Questions:**

-- Is there any characterization on size of m-club set given certain regularity structure of game. This will provide more predictive power about the asymptotic behavior of common learning dynamics

**Limitations:**

Some typographical errors

1. In supplementary material, line 267 has ??
2. In equation C.13 the right hand side is missing.

---

> ### Author Rebuttal · Authors · 2023-08-09
>
> Thank you again for your encouraging input and positive evaluation. We reply to your questions below and we will revise our manuscript accordingly at the next revision opportunity:
>
>
> 1. "*Is there any characterization on size of m-club set given certain regularity structure of game. This will provide more predictive power about the asymptotic behavior of common learning dynamics.*"
>
>    **Reply.** To the best of our knowledge, a complete characterization of (the size of) the support of an m-club set is an open question in the literature. However, under certain structural hypotheses, it is indeed possible to predict what these sets will look like: for example, in (generic) congestion games, the support of any mixed Nash equilibrium contains that of a strict equilibrium, so only strict Nash equilibria can be m-club (a fact which, coupled with Theorem 1, implies that the only irreducibly stable sets of regularized learning in congestion games are strict equilibria). We conjecture that the class of $(\lambda,\mu)$-smooth games introduced by Roughgarden may enjoy similar structural properties, but we are not aware of a specific characterization along these lines.
>
> ---
>
> 2. **Typographical errors**
>    - "*Line 267 has ??*": apologies, this was a broken reference to Appendix C.
>    - "*In equation C.13 the right hand side is missing.*": indeed, (C.13) should read
>    $$
>   \lim_{t\to\infty} \frac{\sum_{s=1}^{t} \gamma_s^2 (1+B_s^2+\sigma_s^2)}{\sum_{s=1}^{t} \gamma_s} = 0
>    $$
> Thanks for spotting these two typos, consider them fixed!
>
> ---
>
> Please let us know if you have any further questions - and thank you again for your input and positive evaluation!
>
> Kind regards,
>
> The authors

---

> > ### Comment · Reviewer_nBFX · 2023-08-19
> > **Acknowledgement**
> >
> > I have read the rebuttal and comments from other authors. I will keep my score.

---

### Author Rebuttal · Authors · 2023-08-09

Dear AC, dear reviewers,

We are sincerely grateful for your time, input and positive evaluation. To streamline the discussion phase, we reply to each reviewer's questions in a separate point-by-point thread below.

Thank you again for your time and positive input. Kind regards,

The authors

---

### Decision · Program_Chairs · 2023-09-21

**Decision:**

Accept (spotlight)

**Comment:**

This paper studies the long-term behavior of regularized, no-regret learning in multiplayer games. Concretely, the paper considers a more stringent criteria---closedness under better replies (CLUB), and shows that a product of pure strategies is CLUB if and only if its span is stable and attracting under regularized learning. Additionally, the rates of convergence of several algorithms are given.

The paper considers a new interesting setting, and achieves novel results. The reviewers are satisfied with author feedback. We thus recommend acceptance.